# S-CLIP: Semi-supervised Vision-Language Learning using Few Specialist Captions

**Sangwoo Mo**[1,2]    **Minkyu Kim**[1,3]    **Kyungmin Lee**[1]    **Jinwoo Shin**[1]
[1]KAIST    [2]University of Michigan    [3]KRAFTON

## Abstract

Vision-language models, such as contrastive language-image pre-training (CLIP), have demonstrated impressive results in natural image domains. However, these models often struggle when applied to specialized domains like remote sensing, and adapting to such domains is challenging due to the limited number of image-text pairs available for training. To address this, we propose S-CLIP, a semi-supervised learning method for training CLIP that utilizes additional unpaired images. S-CLIP employs two pseudo-labeling strategies specifically designed for contrastive learning and the language modality. The caption-level pseudo-label is given by a combination of captions of paired images, obtained by solving an optimal transport problem between unpaired and paired images. The keyword-level pseudo-label is given by a keyword in the caption of the nearest paired image, trained through partial label learning that assumes a candidate set of labels for supervision instead of the exact one. By combining these objectives, S-CLIP significantly enhances the training of CLIP using only a few image-text pairs, as demonstrated in various specialist domains, including remote sensing, fashion, scientific figures, and comics. For instance, S-CLIP improves CLIP by 10% for zero-shot classification and 4% for image-text retrieval on the remote sensing benchmark, matching the performance of supervised CLIP while using three times fewer image-text pairs.[1]

## 1   Introduction

Pre-trained vision-language models have achieved remarkable success, providing a foundation for numerous downstream tasks [1]. However, these models often struggle when applied to specialized domains such as remote sensing or medical imaging [2]. This is because the models are trained on web-crawled data that may not fully capture the diversity and complexity of these domains [3, 4]. To address this issue, previous research has focused on constructing large-scale pre-training datasets for each domain [5, 6]. However, annotating caption for each specialized domain can be expensive and time-consuming, which limits their applicability across various domains.

Several studies have attempted to reduce the number of image-text pairs used for vision-language pre-training. However, these approaches often rely on additional information, such as pre-trained object detectors [7–9], or class-annotated images [10], which may not be applicable to specialized domains. Other approaches leverage self-supervised learning to utilize unpaired data [11–14], but they do not fully exploit the information provided by image-text pairs. Research on leveraging a few image-text pairs to improve vision-language pre-training remains underexplored.

In various areas of machine learning, particularly in image classification, semi-supervised learning [15] is a popular approach for training a model using limited annotations. This technique typically relies on pseudo-labeling [16], which predicts labels for unlabeled data by propagating information

---

*Work done at KAIST. Corresponds to: `swmo@umich.edu`
[1]Code: `https://github.com/alinlab/s-clip`

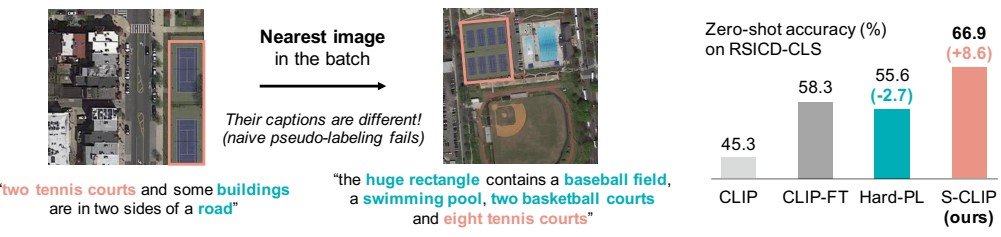

(a) Problem of naive pseudo-labeling (use the nearest caption)      (b) Comparison of models

Figure 1: **Motivation.** (a) Images and captions are from the remote sensing dataset, RSICD. Unlike class labels, captions vary widely across images, making the naive pseudo-labeling of assigning the nearest caption ineffective. (b) Zero-shot accuracy on the RSICD-CLS dataset. Bracket indicates the gap from CLIP fine-tuned on remote sensing datasets, called CLIP-FT. The original CLIP struggles in this specialized domain, and CLIP-FT gives a large gain. As discussed in (a), naive pseudo-labeling (Hard-PL) harms CLIP-FT, while our S-CLIP provides substantial improvement.

from labeled data and refining the model using them. Combining this approach with self-supervised learning [17] has significantly improved classifiers using only a few class labels [18–22].

A natural question is whether semi-supervised learning, particularly pseudo-labeling, can improve vision-language pre-training. However, the techniques from image classification cannot be directly applied to vision-language models since captions are diverse and often uniquely associated with each image. Thus, the naive pseudo-labeling approach of assigning the caption of the nearest labeled image[2] (Hard-PL) [16] can mislead the models. Figure 1a demonstrates that even visually similar images often have different captions, although they may share keywords such as "tennis court." As a result, Figure 1b shows that Hard-PL harms the performance of CLIP fine-tuning. This observation motivates the proper design of pseudo-labeling method that effectively utilizes captions.

**Contribution.** We propose S-CLIP, a novel semi-supervised learning method for vision-language pre-training, particularly CLIP [23]. S-CLIP introduces two novel pseudo-labeling methods specifically designed for contrastive learning and the language modality, as illustrated in Figure 2:

- *Caption-level pseudo-label.* We assume that the semantics of an unlabeled image can be expressed as a combination of those of labeled images. To achieve this, we define the caption-level pseudo-label as a probability distribution over the labeled images, which guides the relationship of an unlabeled image with its corresponding captions. Specifically, the pseudo-labels are obtained by solving an optimal transport [24, 25] problem between the unlabeled and labeled images. This approach prevents pseudo-labels from collapsing to a few captions and ensures robust training, particularly when dealing with distribution shifts between unlabeled and labeled images.

- *Keyword-level pseudo-label.* We assume that an unlabeled image shares keywords with visually similar images, even if their full captions are not identical. Therefore, we define the keyword-level pseudo-label as one of the keywords in the nearest labeled image to an unlabeled image. This approach creates a candidate set of target keywords instead of a single exact one, and the training can be formulated as a partial label learning [26] problem. This loss helps the model understand the local components of unlabeled images, leveraging the structure of language.

We note that both pseudo-labels are complementary. Caption-level pseudo-label helps the model understand the global structure of language, which is more effective for image-text retrieval. Keyword-level pseudo-label helps the model understand the local phrases, which is more effective for zero-shot classification. Combining them achieves the best of both worlds (Section 5.5).

We demonstrate the effectiveness of our method in various specialist domains with limited available image-text pairs, including remote sensing [27], fashion [28], scientific figures [29], and comics [30] domains. In the remote sensing domain, S-CLIP outperforms CLIP fine-tuning and semi-supervised learning competitors in five zero-shot classification and six image-text retrieval tasks. For instance, S-CLIP improves zero-shot accuracy on WHU-RS19 [31] by 10.4% and image-to-text retrieval R@5

---

[2]We use the terms "paired" and "labeled" interchangeably, as well as "unpaired" and "unlabeled," depending on the context of vision-language pre-training or semi-supervised learning.

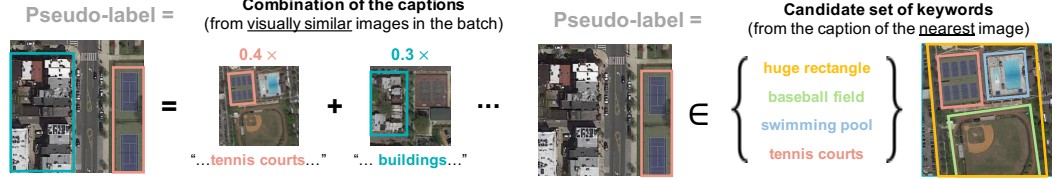

(a) Caption-level pseudo-label          (b) Keyword-level pseudo-label

Figure 2: **Method overview.** Conceptual illustration of our proposed S-CLIP, utilizing (a) caption-level and (b) keyword-level pseudo-labels. In (a), the pseudo-label is given by a probability distribution over the captions, obtained by solving an optimal transport problem between unlabeled and labeled images. In (b), the pseudo-label is given by the candidate set of keywords from the caption of the nearest labeled image, which is learned by the partial label learning algorithm.

on UCM [32] by 4.4% compared to CLIP fine-tuning. S-CLIP remains robust even when unlabeled images are drawn from a different dataset. In the comics domain, the Simpsons [30] dataset only includes 800 image-text pairs. Therefore, we incorporate the Simpsons Characters [33] dataset, which consists of 30,000 images without captions. Here, S-CLIP improves the text→image retrieval R@1 of CLIP from 11.8% to 15.8%, providing a relative gain of 33%.

## 2 Related work

**Vision-language pre-training (VLP).** Pre-training vision-language models has achieved remarkable success, providing a foundation for many downstream tasks [1]. Numerous approaches have been proposed, including reconstructing masked inputs [34–38], learning a joint embedding of vision and language modalities [23, 39–42], generating language descriptions from image inputs [43–45], and connecting pre-trained unimodal image and language models [46–48]. CLIP [23] is a representative model that learns a joint embedding between vision and language modalities. CLIP has demonstrated effectiveness in various applications, including zero-shot classification [23], image-text retrieval [49], open-vocabulary object segmentation [50, 51], and out-of-distribution detection [52].

**VLP for specialist domains.** Despite their success in natural image domains, VLP models often struggle when applied to specialist domains like remote sensing [2]. This is because they are trained on web-crawled data that may not capture the diversity and complexity of these domains [3, 4]. Data in these fields is often scarce, has limited accessibility, and requires expert annotations, making it difficult to obtain the image-text pairs needed for VLP training. While attempts have been made to train VLP models for specialized domains, including medical [10, 53–59], fashion [60–63], and remote sensing [64], previous research has mainly focused on constructing large-scale pre-training datasets for each domain [5, 6], which can be expensive and time-consuming.

**VLP with limited pairs.** Several studies trained VLP models with a limited number of image-text pairs. Hsu et al. [65] used adversarial domain adaptation to regularize unpaired data embeddings similar to paired data. However, this approach can cause problems when the unpaired data is drawn from a different distribution. MedCLIP [10] utilized images with class labels to reduce the need for captions. Nevertheless, it still requires class labels, limiting the use of large unlabeled data. Other studies explored unsupervised or semi-supervised VLP training in image domains [7–9]. However, they rely on a pre-trained object detector to align detected objects with keywords in captions, making them unsuitable for specialist domains. Other works leverage self-supervised learning on each modality [11–14], but do not fully exploit the information provided by image-text pairs. In contrast, our method leverages image-text pairs and is generally applicable without extra information.

**Semi-supervised learning.** Semi-supervised learning [15] aims to train a model using a small labeled dataset and a large unlabeled dataset. Techniques for semi-supervised learning can be categorized into two types: pseudo-labeling (or label propagation) [16, 66], which utilizes information from the small labeled dataset, and self-supervised learning [17], which learns representations from the unlabeled dataset. Combining these techniques has achieved remarkable success in semi-supervised learning, particularly for training classifiers with few class labels [18–22]. Prior works on robust semi-supervised learning focus on scenarios where the distributions of unlabeled and labeled data

shift [67–69]. This issue naturally arises in vision-language pre-training due to the uniqueness of captions for each image. Our method addresses this issue using soft labels and performs robustly even when the unlabeled images are from a different dataset than the labeled images.

## 3 Background

Before we demonstrate our semi-supervised vision-language pre-training method, we provide a brief overview on contrastive language-image pre-training and semi-supervised learning.

### 3.1 Contrastive language-image pre-training

Contrastive language-image pre-training (CLIP) [23] is a popular vision-language model that learns a joint embedding space connecting images $x$ and texts $y$.[3] CLIP is trained using a batch of paired images and texts $\{x_i, y_i\}_{i=1}^N$ through contrastive learning, where those from the same pair are treated as positive and different pairs as negative. The training objective of CLIP involves two classification tasks: predicting a text given an image $p(y|x)$ and predicting an image given a text $p(x|y)$. Labels are assigned to the $N$ samples in the batch, corresponding to their pairs as the target. The embeddings are normalized to have a unit norm, and CLIP predicts the label using a softmax function with a temperature parameter $\tau > 0$, applied to the cosine similarity of the embeddings.

Specifically, we denote the softmax classifier $\sigma_\tau$ as a function of input and target embeddings. This allows us to represent $p(y|x) = \sigma_\tau(x, \{y_i\}_{i=1}^N) \in \mathbb{R}^N$ and $p(x|y) = \sigma_\tau(y, \{x_i\}_{i=1}^N) \in \mathbb{R}^N$, which denote the class probabilities of CLIP. These probabilities are calculated as follows: $p(y = y_i|x) = \exp(x \cdot y_i/\tau)/(\sum_{j=1}^N \exp(x \cdot y_j/\tau))$. The CLIP model minimizes the following objective:

$$\mathcal{L}_{\text{CLIP}} = \frac{1}{2N} \sum_{i=1}^N \left( H(p(y|x_i), \mathbf{e}_i) + H(p(x|y_i), \mathbf{e}_i) \right), \tag{1}$$

where $H$ denotes the cross-entropy loss and $\mathbf{e}_i \in \mathbb{R}^N$ is a one-hot vector with the $i$-th element being one. The CLIP model can be used for zero-shot classification or image-text retrieval by computing cosine similarity in the embedding space. For example, in zero-shot classification, the text embedding of the class prompt that is closest to an image embedding can be identified.

### 3.2 Semi-supervised learning

Semi-supervised learning (Semi-SL) [15] aims to train a model using a small number of labeled data $\{x_i, y_i\}_{i=1}^N$ and a large number of unlabeled data $\{u_i\}_{i=1}^M$, i.e., when $M \gg N$. Various Semi-SL methods have been proposed to enhance neural network performance, particularly for image classification [19–22].[4] These methods typically rely on pseudo-labels [16], obtained by predicting classes for unlabeled data using a model. The model is then refined by minimizing the cross-entropy loss $\frac{1}{M} \sum_{i=1}^M H(p(y|u_i), q_i)$ with respect to the prediction $p(y|u_i) \in \mathbb{R}^C$ and pseudo-label $q_i \in \mathbb{R}^C$ for unlabeled data $u_i$. Pseudo-labels can take the form of a hard label, represented by a one-hot vector identifying the class with the highest probability [16], or a soft label that calibrates this hard label [21]. Pseudo-labels minimize prediction entropy by sharpening specific classes [70].

**Semi-SL for CLIP.** Assigning pseudo-labels in CLIP poses challenges due to the unique captions associated with each image. The assumption that unlabeled data belongs to the same class as labeled data does not hold here. Therefore, we propose a novel Semi-SL method for CLIP.

## 4 S-CLIP: Semi-supervised Vision-Language Pre-training

We extend the training of CLIP to incorporate unpaired images $\{u_i\}_{i=1}^M$ alongside the image-text pairs $\{x_i, y_i\}_{i=1}^N$. To achieve this, we introduce S-CLIP, which integrates two novel pseudo-labeling methods considering contrastive learning and the language modality. The conceptual illustrations of the proposed pseudo-labels and training objectives are shown in Figure 2 and Figure 3, respectively.

---

[3]We use the notations $x$ and $y$ to represent image and text embeddings computed by the CLIP encoders. However, for simplicity, we often omit the term "embeddings" when referring to them.

[4]In image classification, the targets $y$ are represented by one-hot vectors of size $C$ for $C$ classes.

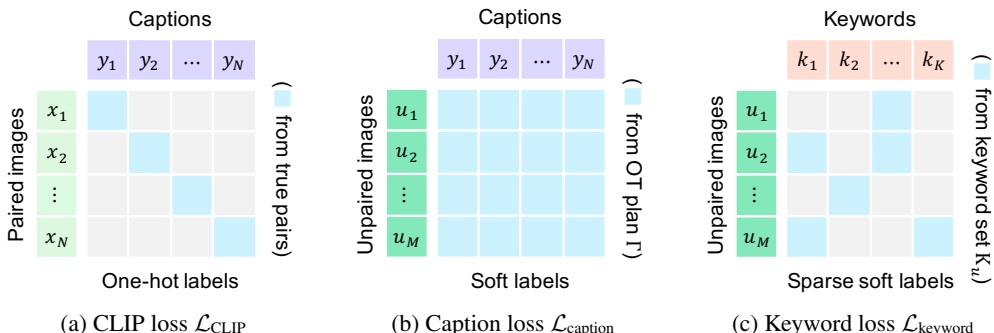

Figure 3: **Training objective.** Conceptual illustrations of the training objectives of S-CLIP: (a) is the original CLIP loss in Eq. (1) using paired images and texts; (b) and (c) are our proposed caption-level and keyword-level pseudo-label losses in Eq. (3) and Eq. (4), respectively, applied to unpaired images. Caption-level pseudo-labels are soft labels obtained from the optimal transport (OT) plan $\Gamma$, while keyword-level pseudo-labels are sparse soft labels obtained from the candidate keyword set $\mathsf{K}_u$.

## 4.1   Caption-level pseudo-label

Although the semantics of an unlabeled image $u$ may not precisely match any captions $y_i$, visually similar images often share similar semantics. Therefore, the caption for unlabeled image $u$ can be expressed as a combination of the captions $\{y_i\}$. To achieve this, we use caption-level pseudo-labels, which represent a probability distribution over the captions. These pseudo-labels serve as the target for the $N$-way classification problem in contrastive learning, with a batch size of $N$. Figure 2a demonstrates how caption-level pseudo-label for an unlabeled image is obtained.

The pseudo-labels are derived from the relationship between unlabeled and labeled images, formulated as an optimal transport (OT) [24] problem. We define the cost function $\mathrm{C} \in \mathbb{R}^{M \times N}$, where $M$ and $N$ represent the number of unlabeled and labeled images, respectively. The cost is given by the negative cosine similarity between the normalized embeddings of the unlabeled image $u_i$ and the labeled image $x_j$, i.e., $\mathrm{C}_{ij} = 1 - u_i \cdot x_j$. Then, we solve following entropic regularized OT [25] problem:

$$\min_{\Gamma \in \Pi(\mathbf{p}, \mathbf{q})} \langle \Gamma, \mathrm{C} \rangle - \lambda H(\Gamma), \quad \Pi(\mathbf{p}, \mathbf{q}) = \{\Gamma \in \mathbb{R}_+^{M \times N} \mid \Gamma \mathbf{1}_N = \mathbf{p}, \ \Gamma^\top \mathbf{1}_M = \mathbf{q}\}. \tag{2}$$

Here, $\Pi(\mathbf{p}, \mathbf{q})$ represents a set of transportation plans that satisfy the flow constraint. The sums of flows from sources and to sinks match the vectors $\mathbf{p} \in \mathbb{R}_+^M$ and $\mathbf{q} \in \mathbb{R}_+^N$, respectively, with both $\mathbf{p}$ and $\mathbf{q}$ adding up to one. We set them as uniform probabilities, and they worked in our experiments, even when unlabeled and labeled data have distribution shifts. $\mathbf{1}_N$ denotes the all-one vector of dimension $N$. The entropic regularizer $H(\Gamma) = -\sum_{i,j} \Gamma_{ij} \log \Gamma_{ij}$ with scale $\lambda$ smooths the transportation plan, enabling an efficient solution using the Sinkhorn-Knopp [25] algorithm.

Once we obtain the transportation plan $\Gamma$, we define the pseudo-label for an unlabeled image $u_i$ by normalizing its plan to sum up to one. Specifically, the $j$-th element of the pseudo-label $q_i \in \mathbb{R}^N$ is computed as $\Gamma_{ij} / \sum_j \Gamma_{ij}$. This pseudo-label is then utilized to guide the probability distribution over $N$ captions, where the prediction $p(y|u_i) \in \mathbb{R}^N$ is determined by a softmax classifier $\sigma_\tau(u_i, \{y_i\}_{i=1}^N)$. The loss function for the caption-level pseudo-label aims to minimize the following:

$$\mathcal{L}_{\text{caption}} = \frac{1}{M} \sum_{i=1}^M H(p(y|u_i), q_i). \tag{3}$$

The prediction $p(y|u_i)$ relates the unlabeled image $u_i$ to the text embeddings $y$, while the pseudo-label $q_i$ relates it to the image embeddings $x$. Consequently, the proposed loss function aligns the relationships between image and text embeddings, as suggested in supervised CLIP [71–73]. In our experiments, we compare the usage of text and image embeddings for pseudo-label computation, and using image embeddings outperformed using text embeddings due to their robustness in novel specialist domains. We also explore computing pseudo-embeddings [9], but it yields inferior results as synthesizing pseudo-embeddings is more challenging than inferring pseudo-labels.

**Effect of OT.** Optimal transport balances the flow between unlabeled and labeled images, ensuring robust pseudo-label assignments without concentration on a few labels. In vision-language tasks, diverse and imbalanced captions often cause image embeddings to focus on a few nearest captions, resulting in the collapse of embeddings. This challenge is magnified when employing semi-supervised learning with distribution shifts [67]. We empirically validate the detrimental effect of naive pseudo-labeling approaches, especially in the presence of distribution shifts.

Compared to soft-nearest neighbor pseudo-labels (Soft-PL) [21], our method achieves balance through Sinkhorn iterations. Specifically, it reverts to Soft-PL when the iteration count is zero. In our experiments, we use 10 iterations, and increasing the iteration count has only a marginal impact on performance. The computational cost is negligible as the process occurs in a low-dimensional embedding space. We set the scale of the entropic regularizer to match the softmax temperature, $\lambda = \tau$. This adjustment scales the cost function by $1/\tau$, aligning it with the cosine similarity scale learned by CLIP. More information on the OT and Soft-PL relationship is in Appendix A.

**OT for vision-language models.** Several works have utilized OT for vision-language models, but for problems other than semi-supervised learning. We discuss these works in Appendix B.

### 4.2 Keyword-level pseudo-label

We propose a keyword-level pseudo-labeling approach to overcome the limitations of caption-level pseudo-labels in capturing the meanings of words in captions. By leveraging the compositionality of language, we assume that visually similar images may share keywords, even if the entire captions differ. This enables us to guide unlabeled images using keywords from their nearest neighbors. It is important to note that an image may have multiple keywords, and visually similar images share only a subset. This aligns with the partial label learning (PLL) [26] problem, where the target label is unknown, and we have access to a candidate set of labels. For instance, in Figure 1a, the image shares the keyword "tennis court" with its nearest neighbor but not "baseball field." Consequently, the nearest neighbor provides a candidate set of keywords, as illustrated in Figure 2b.

To identify candidate keywords for an unlabeled image $u$, we assume a pre-defined set of keywords with corresponding embeddings $\mathsf{K} = \{k_i\}_{i=1}^K$. These keywords can be obtained from class names, if available, or extracted by using algorithms like YAKE [74] applied to captions. Given an unlabeled image $u$, we find the nearest labeled image $x$ using the OT assignments from the previous section. Let $\{c_1, \ldots, c_l\}$ be the indices of keywords present in labeled image $x$, denoted as $\mathsf{K}_u = \{k_{c_1}, \ldots, k_{c_l}\}$. This forms the candidate set of target keywords for $u$. While we simply use the keywords explicitly from the caption, some additional steps can be taken to infer synonyms or related concepts.

In the context of partial label learning, we assume that the ground-truth target label for the unlabeled image $u$ belongs to the candidate set $\mathsf{K}_u$. PLL aims to minimize the loss by selecting the minimum value among candidate labels, $\min_{k \in \mathsf{K}_u} H(p(k|u), k)$, rather than specifying a single target label. However, relying on a single minimum can be risky. Previous works have addressed this by using soft labels for candidates [75, 76]. This involves minimizing $H(p(k|u), q)$, where $q \in \mathbb{R}^K$ is a soft label with positive values for keywords in the candidate set and zero for others.

Similarly, we define the pseudo-label $q_i$ for the unlabeled image $u_i$ based on the similarity between the embeddings of $u_i$ and the keywords $k_j$ in the candidate set $\mathsf{K}_u$. Specifically, the $j$-th element of the pseudo-label $q_i$ is calculated using a softmax function $\sigma_\tau(u_i, \mathsf{K}_u)$ for $k \in \mathsf{K}_u$ and is set to zero otherwise. This pseudo-label is then utilized to guide the probability distribution over $K$ keywords, where the prediction $p(k|u) \in \mathbb{R}^K$ is determined by a softmax classifier $\sigma_\tau(u_i, \mathsf{K})$. The loss function for the keyword-level pseudo-label aims to minimize the following:

$$\mathcal{L}_{\text{keyword}} = \frac{1}{M} \sum_{i=1}^M H(p(k|u_i), q_i). \tag{4}$$

Both the prediction $p(k|u_i)$ and the pseudo-label $q_i$ are related to the unlabeled image and keywords. However, the restricted candidate set effectively guides the learning process through PLL.

**Training objective.** We apply the CLIP loss to paired images and texts and the proposed pseudo-label losses to unpaired images. The pseudo-label losses are halved to match the scale of the image part in the CLIP loss. In summary, our final training objective becomes $\mathcal{L}_{\text{CLIP}} + 1/2 \left( \mathcal{L}_{\text{caption}} + \mathcal{L}_{\text{keyword}} \right)$.

Table 1: Zero-shot classification results on remote sensing datasets. We compare the original CLIP, supervised CLIP fine-tuned on labeled data (L), and semi-supervised methods that utilize unlabeled data sampled from the same (L=U) or different (L≠U) distribution as the labeled data. Parentheses indicate the performance gap from the supervised CLIP, where values highlighted in green indicate gaps larger than one. Bolds denote the best results among the semi-supervised methods within the same setups. S-CLIP consistently improves zero-shot accuracy, even when the unlabeled data has a distribution shift. In contrast, naive pseudo-labeling often harms the accuracy.

| Method | Data | RSICD-CLS | UCM-CLS | WHU-RS19 | RSSCN7 | AID |
|---|---|---|---|---|---|---|
| CLIP (original) | - | 45.3 | 50.5 | 65.5 | 58.9 | 47.8 |
| CLIP (fine-tune) | L | $58.3_{\pm0.3}$ | $63.5_{\pm3.4}$ | $76.5_{\pm3.2}$ | $61.9_{\pm1.2}$ | $63.1_{\pm1.3}$ |
| Hard-PL [16] | | $56.6_{\pm3.5}$ (-1.7) | $61.6_{\pm2.2}$ (-1.9) | $78.1_{\pm2.5}$ (+1.6) | $63.9_{\pm2.1}$ (+2.0) | $63.2_{\pm2.6}$ (+0.1) |
| Soft-PL [21] | L=U | $62.5_{\pm0.8}$ (+4.2) | $65.7_{\pm2.7}$ (+2.2) | $83.7_{\pm2.7}$ (+7.2) | $65.7_{\pm0.6}$ (+3.8) | $68.0_{\pm0.7}$ (+4.9) |
| S-CLIP (ours) | | $\mathbf{66.9}_{\pm1.7}$ (+8.6) | $\mathbf{66.7}_{\pm1.6}$ (+3.2) | $\mathbf{86.9}_{\pm2.0}$ (+10.4) | $\mathbf{66.2}_{\pm1.1}$ (+4.3) | $\mathbf{73.0}_{\pm0.3}$ (+9.9) |
| Hard-PL [16] | | $55.8_{\pm1.2}$ (-2.5) | $61.0_{\pm6.2}$ (-2.5) | $76.3_{\pm1.1}$ (-0.2) | $62.5_{\pm2.7}$ (+0.6) | $62.3_{\pm1.4}$ (-0.8) |
| Soft-PL [21] | L≠U | $56.1_{\pm3.5}$ (-2.2) | $62.4_{\pm1.8}$ (-1.1) | $79.5_{\pm3.6}$ (+3.0) | $63.3_{\pm2.6}$ (+1.4) | $62.4_{\pm2.1}$ (-0.7) |
| S-CLIP (ours) | | $\mathbf{65.3}_{\pm2.1}$ (+7.0) | $\mathbf{66.0}_{\pm0.4}$ (+2.5) | $\mathbf{86.3}_{\pm1.1}$ (+9.8) | $\mathbf{68.3}_{\pm2.2}$ (+6.4) | $\mathbf{70.8}_{\pm2.5}$ (+7.7) |

## 5   Experiments

We demonstrate the effectiveness of our method in various specialist domains with limited available image-text pairs, including remote sensing, fashion, scientific figures, and comics domains.

**Setup.** We use a pre-trained CLIP model on natural images [23], called CLIP (original). However, this model often struggles in specialist domains. Therefore, we fine-tune the model with domain-specific image-text pairs, called CLIP (fine-tune). We compare this supervised model with semi-supervised methods using unpaired images. We use a batch size of 64 per GPU, with a total of 4 GPUs. To ensure fair GPU memory usage in semi-supervised learning, we employ 32 image-caption pairs and 32 unpaired images for each mini-batch. Models are evaluated on zero-shot classification and image-text retrieval tasks, measuring Top-1 classification accuracy (%) and recall at K (R@K). We report the average and standard deviation across three random seeds. We follow the training recipe of OpenClip [77] if not specified. Additional experimental details are in Appendix C.

### 5.1   Baselines

Our proposed S-CLIP includes two novel pseudo-labeling methods. Thus, we mainly compare S-CLIP with other pseudo-labeling approaches, which are extensions of semi-supervised classification techniques. Our baselines, Hard-PL and Soft-PL, extend the hard (top-1) and soft nearest neighbor approaches for pseudo-label assignment. We then utilize these methods as caption-level pseudo-labels, similar to Eq. (3). The specific forms of each method are as follows.

**Hard-PL.** Given an unlabeled image $u_i$, Hard-PL identifies the top-1 nearest labeled image[5] $x_{i^*}$ based on their cosine similarities, calculated as $i^* = \arg\max_j(u_i \cdot x_j)$. Subsequently, the pseudo-label of $u_i$ is determined by the corresponding caption, denoted as $q_i = y_{i^*}$. This pseudo-label serves as the target for predicting $p(y|u_i)$, and the loss is computed as $\sum_i H(p(y|u_i), q_i)$.

**Soft-PL.** Soft-PL calibrates the pseudo-label of Hard-PL by considering all the relations between labeled and unlabeled images, known as soft nearest neighbor [21], instead of selecting only the top-1 nearest image. For an unlabeled image $u_i$ and a set of labeled images $\{x_i\}$, these relations are determined by applying softmax to their cosine similarities. The pseudo-label is defined as $q_i = \sigma_\tau(u_i, \{x_i\}_{i=1}^N) \in \mathbb{R}^N$, where the pseudo-label has a higher value close to one for visually similar images. We use the same softmax temperature $\tau$ as CLIP, which is used to compute the similarities between the labeled image and text embeddings. The loss is computed as $\sum_i H(p(y|u_i), q_i)$.

### 5.2   Remote sensing datasets

We train vision-language models using the union of RSICD [27], UCM [32], and Sydney [78], named RS-ALL, following the setup of [64]. For semi-supervised learning, we subsample 10% of image-text

---

[5]We also tested the version directly finding the nearest text for both Hard-PL and Soft-PL. However, the similarities in visual embeddings give more robust results, as discussed in the ablation study.

Table 2: Image-text retrieval results on remote sensing datasets, following the same setup of Table 1. Bolds denote the best results among the semi-supervised methods within the same setups. S-CLIP performs best in most cases, except the supervised baseline works best for text→image retrieval on the Sydney dataset. Naive pseudo-labeling also helps, but the gain is often unstable.

| Method | Data | Image→text R@5 | | | Text→image R@5 | | |
|---|---|---|---|---|---|---|---|
| | | RSICD | UCM | Sydney | RSICD | UCM | Sydney |
| CLIP (original) | - | 9.4 | 34.3 | 36.2 | 10.1 | 24.8 | 51.7 |
| CLIP (fine-tune) | L | $15.4_{\pm1.7}$ | $41.3_{\pm1.8}$ | $47.1_{\pm6.5}$ | $15.1_{\pm1.0}$ | $40.9_{\pm1.6}$ | $56.1_{\pm2.4}$ |
| Hard-PL [16] | | $16.1_{\pm0.2}$ | $40.8_{\pm2.9}$ | $43.1_{\pm3.0}$ | $15.7_{\pm0.7}$ | $40.5_{\pm3.0}$ | $47.7_{\pm5.3}$ |
| Soft-PL [21] | L=U | $17.0_{\pm0.9}$ | $43.2_{\pm3.9}$ | $42.0_{\pm4.3}$ | $16.5_{\pm0.1}$ | $42.9_{\pm3.3}$ | $50.2_{\pm4.9}$ |
| S-CLIP (ours) | | $\mathbf{18.4}_{\pm0.6}$ | $\mathbf{45.7}_{\pm1.4}$ | $\mathbf{50.0}_{\pm3.0}$ | $\mathbf{16.8}_{\pm1.2}$ | $\mathbf{43.5}_{\pm1.5}$ | $\mathbf{55.1}_{\pm2.0}$ |
| Hard-PL [16] | | $15.8_{\pm1.0}$ | $42.1_{\pm5.7}$ | $47.1_{\pm4.0}$ | $15.3_{\pm0.4}$ | $42.1_{\pm3.1}$ | $50.0_{\pm1.7}$ |
| Soft-PL [21] | L≠U | $16.5_{\pm1.1}$ | $40.2_{\pm3.2}$ | $46.6_{\pm4.6}$ | $15.4_{\pm0.2}$ | $40.8_{\pm0.5}$ | $\mathbf{54.0}_{\pm4.0}$ |
| S-CLIP (ours) | | $\mathbf{17.1}_{\pm0.8}$ | $\mathbf{43.5}_{\pm3.5}$ | $\mathbf{48.9}_{\pm2.6}$ | $\mathbf{15.8}_{\pm0.7}$ | $\mathbf{42.5}_{\pm1.5}$ | $52.3_{\pm1.0}$ |

pairs as labeled data (L), while the remaining 90% of images (L=U) or unlabeled images from the RESISC45 [79] dataset (L≠U) served as unlabeled data. In Appendix D, we demonstrate that our method consistently gives improvements over CLIP across various setups. These include different neural architectures, varied ratios of image-text pairs, and even scenarios where we use all image-text pairs from RS-ALL in conjunction with unlabeled images from RESISC45.

For zero-shot classification, we use the validation sets of the classification versions of the RSICD and UCM datasets, denoted as RSCID-CLS and UCM-CLS, respectively. To evaluate the generalization abilities, we test models on unseen datasets such as WHU-RS19 [31], RSSCN7 [80], and AID [81]. For image-text retrieval, we use the validation sets of the RSICD, UCM, and Sydney datasets.

**Zero-shot classification.** Table 1 presents the zero-shot classification results. S-CLIP consistently and significantly outperforms all supervised CLIP fine-tuning and semi-supervised methods. One can make several observations. Firstly, fine-tuning CLIP improves performance over the original CLIP, highlighting the importance of adapting models to the target specialist domain. Secondly, our proposed pseudo-labeling techniques are crucial, as naive pseudo-labeling often harms accuracy. Thirdly, semi-supervised learning methods improve performance on unseen datasets, such as RSSCN7, demonstrating the usefulness of observing more unlabeled images in generalization. Lastly, our method is robust to the distribution shifts (L≠U) setup, which leverages an external source of unlabeled images, a common scenario in practice. In contrast, Soft-PL improves performance in the same distribution (L=U) setup but fails on the distribution shifts scenario, confirming the effectiveness of optimal transport in balancing the assignments of pseudo-labels.

**Image-text retrieval.** Table 2 presents the image-text retrieval results. S-CLIP consistently improves both image→text and text→image retrieval, except in one case where the supervised baseline performs the best. This confirms that S-CLIP also benefits fine language understanding for retrieval. The trend is consistent across evaluation metrics, as seen in the R@1 results in the appendix.

## 5.3 Fashion datasets

We train vision-language models using the union of Fashion200k [28], FashionGen [82], and Polyvore Outfits [83] datasets. We subsample 10% of the image-text pairs as labeled data and the remaining 90% of images as unlabeled data. We evaluate the zero-shot accuracy on the validation sets of all three datasets, considering both super-class and sub-class accuracies for Fashion200k and FashionGen. The class names in Polyvore match the same level of super-class as those in the other datasets.

Table 3 presents the zero-shot classification results. S-CLIP consistently outperforms all supervised and semi-supervised methods. The improvement is more significant for super-class classification but less pronounced for sub-class classification and image-text retrieval. This is because the current pseudo-labeling approach assumes that the semantics of unlabeled image can be represented by the captions in a batch. However, this assumption may not hold true for fine-grained semantics, as certain fine-grained captions may be missing from the batch. Therefore, our method could be further enhanced by increasing the batch size or incorporating a queue of caption embeddings [84, 85].

Table 3: Zero-shot classification results on fashion datasets. Parentheses indicate the performance gap from the supervised CLIP, where values highlighted in green indicate gaps larger than one. Bolds denote the best results. S-CLIP consistently outperforms all supervised and semi-supervised methods.

| | Fashion200k | | FashionGen | | Polyvore |
| Method | Super-class | Sub-class | Super-class | Sub-class | Class |
|---|---|---|---|---|---|
| CLIP (original) | 73.4 | 29.3 | 35.9 | 22.1 | 58.4 |
| CLIP (fine-tune) | $79.4_{\pm1.5}$ | $38.4_{\pm1.5}$ | $41.5_{\pm1.4}$ | $33.1_{\pm1.9}$ | $64.2_{\pm3.1}$ |
| Hard-PL [16] | $74.6_{\pm2.7}$ (-4.8) | $35.0_{\pm2.6}$ (-3.4) | $36.6_{\pm3.9}$ (-4.9) | $28.8_{\pm0.8}$ (-4.3) | $65.5_{\pm0.7}$ (+1.3) |
| Soft-PL [21] | $80.0_{\pm1.1}$ (+0.6) | $37.1_{\pm0.7}$ (-1.3) | $45.5_{\pm3.2}$ (+4.0) | $33.2_{\pm0.3}$ (+0.1) | $73.6_{\pm7.9}$ (+9.4) |
| S-CLIP (ours) | $\mathbf{82.0}_{\pm2.8}$ (+2.6) | $\mathbf{39.5}_{\pm1.1}$ (+1.1) | $\mathbf{55.3}_{\pm2.6}$ (+13.8) | $\mathbf{38.7}_{\pm1.9}$ (+5.6) | $\mathbf{74.6}_{\pm1.8}$ (+10.4) |

Table 4: Image-text retrieval results on SciCap (scientific figures) and Simpsons (comics) datasets. We train the model on each dataset and evaluate on their validation sets. Bolds denote the best results. S-CLIP consistently outperforms all supervised and semi-supervised methods.

| | SciCap | | | | Simpsons | | | |
| | Image→text | | Text→image | | Image→text | | Text→image | |
| Method | R@1 | R@5 | R@1 | R@5 | R@1 | R@5 | R@1 | R@5 |
|---|---|---|---|---|---|---|---|---|
| CLIP (original) | 9.2 | 14.7 | 9.0 | 14.5 | 15.8 | 40.8 | 10.5 | 27.6 |
| CLIP (fine-tune) | $10.5_{\pm0.2}$ | $17.9_{\pm0.2}$ | $10.2_{\pm0.1}$ | $17.2_{\pm0.2}$ | $16.7_{\pm2.0}$ | $41.4_{\pm1.1}$ | $11.8_{\pm2.3}$ | $41.7_{\pm1.5}$ |
| Hard-PL [16] | $9.7_{\pm0.3}$ | $16.8_{\pm0.6}$ | $9.2_{\pm0.6}$ | $16.2_{\pm0.7}$ | $15.4_{\pm1.5}$ | $41.7_{\pm3.3}$ | $12.7_{\pm1.5}$ | $40.8_{\pm2.3}$ |
| Soft-PL [21] | $9.1_{\pm0.5}$ | $15.5_{\pm1.0}$ | $9.7_{\pm0.3}$ | $16.8_{\pm0.5}$ | $15.4_{\pm2.0}$ | $39.9_{\pm2.7}$ | $12.9_{\pm1.9}$ | $36.4_{\pm2.7}$ |
| S-CLIP (ours) | $\mathbf{10.9}_{\pm0.3}$ | $\mathbf{18.8}_{\pm0.3}$ | $\mathbf{11.1}_{\pm0.3}$ | $\mathbf{18.8}_{\pm0.4}$ | $\mathbf{18.4}_{\pm2.3}$ | $\mathbf{43.9}_{\pm1.5}$ | $\mathbf{15.8}_{\pm1.3}$ | $\mathbf{42.1}_{\pm0.1}$ |

## 5.4 More captioning datasets

We conduct experiments on two more specialist domains, namely science figures and comics. For this purpose, we use the SciCap [29] and Simpsons [30] datasets. We train the models on each dataset and evaluate their image-text retrieval performance on the validation sets. The results are presented in Table 4, and detailed discussions can be found in the following paragraphs.

**SciCap.** We subsample 10% of the image-text pairs as labeled data and the remaining 90% of images as unlabeled data. S-CLIP improves upon CLIP, even for the scientific figures domain, which exhibits a substantial gap from the natural image domain that CLIP is pre-trained on. This confirms the practical impact of S-CLIP in extending the applicability of CLIP to various specialist domains.

**Simpsons.** The Simpsons dataset contains only 800 image-text pairs, so we incorporate unlabeled images from another dataset called Simpsons Characters [33]. S-CLIP significantly improves CLIP, even in the presence of distribution shifts between unlabeled and labeled images. For example, it boosts the text→image retrieval R@1 from 11.8% to 15.8% (+33%). This confirms the practical impact of S-CLIP in specialist domains with limited available image-text pairs.

## 5.5 Ablation studies

We conduct an ablation study on the design choices for S-CLIP. The results are reported in Table 5, which compares the average values of zero-shot accuracy (across five tasks in Table 1) and image-text retrieval R@1 (across six tasks in Table 2) for each design. The observations are as follows:

(a) *Training objective.* Caption-level pseudo-labels are more beneficial for image-text retrieval, while keyword-level pseudo-labels are better for zero-shot classification. Intuitively, keyword-level pseudo-labels help the model understand specific words, while caption-level pseudo-labels aid in fine-grained language comprehension. Combining both produces the best results.

(b) *Sinkhorn iteration.* Sinkhorn iterations significantly improve Soft-PL (= Iter. 0) by balancing pseudo-labels. However, it saturates after 10 iterations, so we have used it as the default in our experiments. Increasing the number of iterations ensures a more even distribution of pseudo-labels. For example, the pseudo-labels from Soft-PL often collapse to some index in the batch. Sinkhorn iterations effectively regularize this behavior, reducing the maximum confidence of

Table 5: Ablation study on design choices. We report the average values of zero-shot accuracy (left) and image↔text retrieval (right). Bolds denote the best results. See Section 5.5 for discussions.

| (a) Training objective | | |
| --- | --- | --- |
| CLIP (fine-tune) | 64.7 | 9.3 |
| + Caption-level | 69.6 | 10.5 |
| + Keyword-level | 70.0 | 9.9 |
| + Both | **71.9** | **10.6** |

| (b) Sinkhorn iteration | | |
| --- | --- | --- |
| Iter. 0 | 70.6 | 9.2 |
| Iter. 1 | 71.4 | 9.4 |
| Iter. 5 | 71.6 | 10.2 |
| Iter. 10 | **71.9** | **10.6** |

| (c) Choice of keywords | | |
| --- | --- | --- |
| YAKE-100 | 70.0 | 9.8 |
| YAKE-200 | **72.2** | 9.3 |
| YAKE-300 | 70.6 | 10.2 |
| Class name | 71.9 | **10.6** |

| (d) Pseudo-target | | |
| --- | --- | --- |
| Pseudo-embed | 68.8 | 9.2 |
| Pseudo-label | **71.9** | **10.6** |

| (e) Embedding for OT | | |
| --- | --- | --- |
| Text | 71.6 | 9.7 |
| Image | **71.9** | **10.6** |

| (f) Partial label loss | | |
| --- | --- | --- |
| Hard-max | 66.1 | 10.2 |
| Soft-max | **71.9** | **10.6** |

prediction from 100% to 40%. This regularization is more important in the L$\neq$U setup, where Soft-PL often harms the performance, while ours consistently provides improvement.

(c) *Choice of keywords.* We compare class names and YAKE-$K$, which implies that $K$ keywords are extracted using the YAKE [74] algorithm. Both types of keywords work well, although class names provide more consistent improvement. YAKE provides both nouns, such as "building," and adjectives like "green," "many," or "large," in contrast to the class names, which consist only of nouns. This enables YAKE to understand diverse semantics beyond object recognition, making keyword-level pseudo-labeling applicable to any domain. This applicability is further confirmed on the SciCap and Simpsons datasets, where class names are unavailable.

(d) *Pseudo-target.* Given a pseudo-label $q_i \in \mathbb{R}^N$ for an unlabeled image $u_i$, one can synthesize a pseudo-embedding $z_i = \sum_j q_{ij} \cdot y_j \in \mathbb{R}^d$, where $q_{ij}$ denotes the $j$-th element of $q_i$, and $d$ is the dimension of the embeddings $y_j$. Then, one can consider $\{u_i, z_i\}$ as paired data and apply contrastive learning to it in addition to the true pairs $\{x_i, y_i\}$. However, we found that pseudo-embedding is less effective than pseudo-labeling. This is because accurately estimating the pseudo-embedding is harder than assigning soft probabilities to the true embeddings. Additionally, the pseudo-embeddings lie on an interpolation of true embeddings, which can introduce confusing negatives and potentially harm the effectiveness of contrastive learning.

(e) *Embedding for OT.* We compute pseudo-labels by examining the relationship between unlabeled images and either labeled images or labeled text directly. The relationship with labeled images proves to be more effective in assigning pseudo-labels. Specifically, using text embeddings directly also works reasonably well in the L=U setup. However, it often fails entirely in the L$\neq$U setup. This is because the corresponding captions for unlabeled images are unknown. In contrast, visual similarities can be robustly computed even when the images are unseen.

(f) *Partial label loss.* We use the soft label given by $\sigma_\tau(u_i, \mathsf{K}_u)$ as the target for PLL. We also test the hard version, using the top-1 similar keyword from the candidate set, i.e., $i^* = \arg\max_j(u_i \cdot k_j)$. However, it performs worse, aligning with the observations from prior PLL literature.

We provide additional ablation studies in Appendix E, including results in the general image domain, additional pseudo-labeling baselines, qualitative examples of pseudo-labels, training curves, regularized fine-tuning approaches, selection of mini-batches during training, additional baselines using keyword information, and statistics of keywords in captions.

## 6    Conclusion

We propose S-CLIP, a semi-supervised extension of CLIP that leverages unpaired images to train in specialist domains with limited image-text pairs. We demonstrate its superiority across various domains such as remote sensing, fashion, scientific figures, and comics. We hope that S-CLIP expands the applicability of CLIP. Limitations and broader impacts are discussed in Appendix F.

## Acknowledgements

This work was supported by Institute of Information & communications Technology Planning & Evaluation (IITP) grant funded by the Korea government (MSIT) (No.2019-0-00075, Artificial

Intelligence Graduate School Program (KAIST); No.2021-0-02068, Artificial Intelligence Innovation Hub; No.2022-0-00959, Few-shot Learning of Casual Inference in Vision and Language for Decision Making).

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

# A    Relationship between OT and Soft-PL

The entropic-regularized optimal transport problem described in Eq. (2) can be efficiently solved using the Sinkhorn-Knopp [25] algorithm. For our case, the cost function $C \in \mathbb{R}^{M \times N}$ is given by the negative cosine similarity: $C_{ij} = 1 - u_i \cdot x_j$. The Sinkhorn-Knopp algorithm proceeds as follows: first, it computes the exponentiated cost matrix $K = \exp(\frac{C}{-\lambda})$, where $\lambda$ denotes the scale of the entropic regularizer. Then, the algorithm iteratively updates two vectors: $\mathbf{u} \in \mathbb{R}^{M \times 1}$ and $\mathbf{v} \in \mathbb{R}^{1 \times N}$, which are initialized with uniform probability. The transportation plan $\Gamma$ can be estimated using the exponentiated cost matrix $K$ and the current vectors $\mathbf{u}$ and $\mathbf{v}$. Specifically, it is given by $\Gamma = \mathbf{u} \odot K \odot \mathbf{v}$, where $\odot$ denotes element-wise multiplication.

At the initialization zero, the transportation plan $\Gamma$ is obtained by normalizing the exponentiated cost matrix $K$ since the vectors $\mathbf{u}$ and $\mathbf{v}$ are uniform. Consequently, our caption-level pseudo-label $q_i$ is given by $q_i = K_{ij} / \sum_j K_{ij}$, which is equivalent to the Soft-PL $\sigma_\lambda(u_i, x_i)$ except for the softmax temperature, which is denoted as $\lambda$. This observation motivates us to set $\lambda = \tau$, matching the same temperature as CLIP, for computing the relation between embeddings. From this perspective, our proposed OT approach enhances the robustness of Soft-PL in pseudo-label assignment, especially in cases with distribution shifts between unlabeled and labeled images. Our ablation study demonstrates that increasing the number of Sinkhorn iterations improves the training process.

After initialization, the vectors $\mathbf{u}$ and $\mathbf{v}$ are updated to align with the exponentiated cost matrix $K$ and satisfy the flow constraints $\mathbf{p}$ and $\mathbf{q}$. Specifically, given the current vectors $\mathbf{u}^{(t)}$ and $\mathbf{v}^{(t)}$, the next iteration of vectors is obtained as $\mathbf{u}^{(t+1)} = \frac{\mathbf{p}}{K\mathbf{v}^{(t)}}$ and $\mathbf{v}^{(t+1)} = \frac{\mathbf{q}}{K^\top \mathbf{u}^{(t+1)}}$. This iterative process converges to the optimal solution. Our ablation study demonstrates that a small number of iterations is sufficiently effective, and we employ 10 iterations in our experiments.

# B    Additional related work

**Optimal transport for vision-language models.** Several prior works have applied optimal transport to vision-language models. However, most of them focus on transportation between an image and a caption, connecting objects and phrases [35, 36, 86–88]. This is known as weakly supervised phrase grounding [89], and it assumes a supervised setting with image-text pairs. OTTER [90] calculates transportation between images and text, which aligns with our approach. However, they also assume a supervised setting and focus on calibrating the one-hot target of CLIP. To our knowledge, our work is the first to apply optimal transport for semi-supervised CLIP training.

**Optimal transport for semi-supervised learning.** Some prior works have utilized optimal transport for semi-supervised learning, specifically in image classification [91, 92]. These studies demonstrate the benefits of optimal transport in pseudo-labeling class labels. However, we argue that OT plays a more critical role in vision-language models due to inherent distribution shifts between labeled and unlabeled image captions. Our experiments reveal that naive pseudo-labeling often has detrimental effects, while OT significantly improves semi-supervised CLIP training.

**Partial label learning.** Partial label learning (PLL) [26] relaxes the classification problem by using a candidate set as the target instead of the exact one. This involves optimizing for the best candidate within the set. Instead of determining the hard assignment of the candidate, PRODEN [75] computes soft labels for the elements in the set to smooth the optimization process. PiCO [76] further enhances this approach by incorporating contrastive learning. Our work extends the techniques of PLL to a novel problem: keyword-level pseudo-labeling for vision-language models.

**Label propagation in contrastive learning.** The concept of pseudo-labeling (or label propagation) is also applied in image-only contrastive learning. In addition to using the same image with different augmentations [84, 85], some studies explore the utilization of neighboring images as positive pairs [93–95]. Similar concept can be extended to supervised [96] and semi-supervised [22, 97] contrastive learning, where images sharing the same class label are considered positive pairs.

# C  Experimental details

## C.1  Dataset details

**Remote sensing.** We use the union of RSICD [27], UCM [32], and Sydney [78] datasets for training. The captions for the UCM and Sydney datasets were annotated by Qu et al. [98]. Figure 4 displays the example images and captions, while Table 6 presents the statistics of each dataset. Since an image may have multiple associated captions, we randomly select one for each iteration.

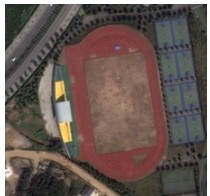 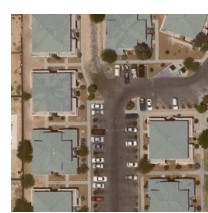 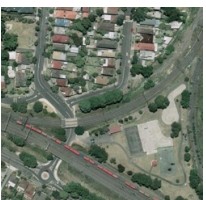

"a playground with eight basketball fields near it is surrounded by many green trees"

"many houses arranged neatly and some roads crossed them with some plants in the roadside"

"a residential area with houses arranged neatly and some roads go through this area"

(a) RSICD

(b) UCM

(c) Sydney

Figure 4: Examples of image-text pairs in the remote sensing domain.

Table 6: Statistics of image-text paired datasets in the remote sensing domain.

|  | RSICD | UCM | Sydney |
|---|---|---|---|
| # of pairs | 8,734 | 1,680 | 497 |

We use the validation sets of RSCID-CLS [27], UCM-CLS [32], WHU-RS19 [31], RSSCN7 [80], and AID [81] datasets for zero-shot classification. RSCID-CLS and UCM-CLS are the classification versions of RSCID and UCM, respectively. Table 7 presents the statistics of each dataset.

Table 7: Statistics of classification datasets in the remote sensing domain.

|  | RSICD-CLS | UCM-CLS | WHU-RS19 | RSSCN7 | AID |
|---|---|---|---|---|---|
| # of images | 1,094 | 2,100 | 1,005 | 2,800 | 10,000 |
| # of classes | 31 | 21 | 19 | 7 | 30 |

**Fashion.** We use the union of Fashion200k [28], FashionGen [82], and Polyvore Outfits [83] datasets for training. Figure 5 displays the example images and captions, while Table 8 presents the statistics of each dataset. Since an item may have multiple associated views of images corresponding to the caption, we randomly select one view for each iteration. For captions, we concatenate the titles and descriptions from FashionGen and Polyvore, and use the sentences from Fashion200k.

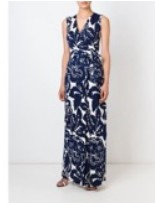 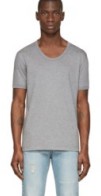 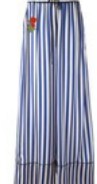

"white printed silk-jersey wrap dress midnight blue"

"grey classic u-neck t-shirt. short sleeve t-shirt in heather grey. tonal logo flag at u-neck collar."

"off-white striped flared pants. multicoloured cotton and silk striped flared pants from off-white."

(a) Fashion200k

(b) FashionGen

(c) Polyvore

Figure 5: Examples of image-text pairs in the fashion domain.

Table 8: Statistics of image-text paired datasets in the fashion domain.

|  | Fashion200k | FashionGen | Polyvore |
|---|---|---|---|
| # of pairs | 61,753 | 60,147 | 71,967 |

We use the validation sets of Fashion200k, FashionGen, and Polyvore datasets for zero-shot classification. Table 9 presents the statistics of each dataset. For Fashion200k and FashionGen, we report both the super-class and the sub-class. The class names in Polyvore match the same level of the super-class as those in the other datasets. For example, the super-class contains class names such as "dress," while the sub-class contains class names such as "casual and day dress."

Table 9: Statistics of classification datasets in the fashion domain.

|  | Fashion200k | | FashionGen | | Polyvore |
|---|---|---|---|---|---|
|  | Super-class | Sub-class | Super-class | Sub-class | Class |
| # of images | 29,785 | | 32,528 | | 14,657 |
| # of classes | 5 | 31 | 48 | 121 | 11 |

**Scientific figures.** We use the SciCap [29] dataset for training. Figure 6 displays the example images and captions, while Table 10 presents the statistics of each dataset. We use figures that do not have subfigures, denoted as the "SciCap-No-Subfig-Img" subset, for simplicity.

**Comics.** We use the Simpsons [30] dataset as labeled data and the Simpsons Character [33] dataset as unlabeled data. Figure 6 displays the example images and captions, while Table 10 presents the statistics of each dataset. We use 90% of pairs from the Simpsons dataset as the training split and the remaining 10% as the validation split. Since the training split contains only 720 image-text pairs, it is necessary to incorporate images from different datasets like Simpsons Characters.

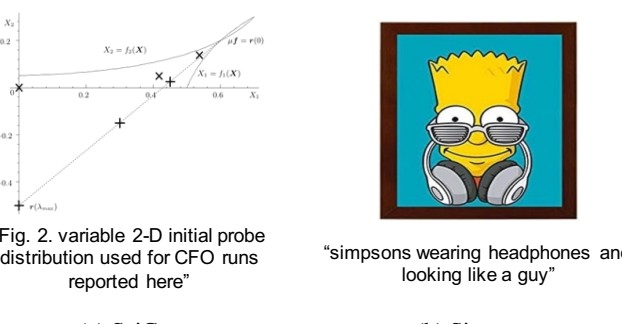

"Fig. 2. variable 2-D initial probe distribution used for CFO runs reported here"

"simpsons wearing headphones and looking like a guy"

(a) SciCap

(b) Simpsons

Figure 6: Examples of image-text pairs in the scientific figure and comics domains.

Table 10: Statistics of image-text paired datasets in the scientific figure and comics domains.

|  | SciCap | Simpsons |
|---|---|---|
| # of pairs | 106,834 | 720 |

## C.2    Implementation details

**Training.** We utilize the OpenCLIP [77] library to implement our algorithms. By default, we follow the original training configuration. We use a batch size of 64 per GPU, with a total of 4 GPUs. The learning rate is set to 5e-5, and we apply the default cosine learning rate scheduling with a warmup period of the first 10 steps. We train all models until the performance saturate, which can vary over the number of image-text pairs. Specifically, for remote sensing, we train models for 25 epochs, for fashion 10 epochs, for scientific figures 5 epochs, and for comics datasets 10 epochs. We run three trials and report the average and standard deviation for all experiments.

We add the supervised CLIP loss and pseudo-label losses for semi-supervised methods. We halve the scale of pseudo-label losses to match the scale of the image part in the CLIP loss. We set the scale of the entropic regularizer to be the same as the softmax temperature of CLIP, denoted as $= \tau$, and set the number of Sinkhorn iterations to 10. Note that our method does not introduce any additional hyperparameters, except for the selection of keyword candidate sets discussed below.

**Keywords.** We use the available class names for remote sensing and fashion datasets while extracting the keywords using the YAKE [74] algorithm from the captions for the SciCap and Simpsons datasets. For remote sensing, we combine the class names from RSICD-CLS and UCM-CLS to obtain 45 keywords, of which around 30 words appeared in the captions. For fashion, we combine the super-class names from Fashion200k and FashionGen to obtain 56 keywords, which are more effective than sub-class names since the captions often do not contain the exact sub-class names.

Our ablation study presents that the keywords extracted by YAKE are also effective, even outperforming the class names in some cases. In the case of remote sensing, the YAKE keywords contain not only nouns like "building" but also adjectives like "green," "many," or "large," which help the model understand the fine-grained information of the scene. Moreover, we utilize YAKE-100 keywords for SciCap and Simpsons when class names are unavailable, and this approach has proven to be effective. Thus, we believe that a reasonable choice of keywords would firmly benefit our method.

**Computation.** The computation bottleneck mostly lies in computing embeddings rather than the loss function. Thus, all methods require a similar amount of time for a single training iteration, except for S-CLIP, which takes longer due to the additional forwarding of keywords. Training duration varies depending on the number of image-text pairs, taking a few minutes to a few hours. Semi-supervised methods take twice as long as supervised CLIP since we draw batches of labeled and unlabeled data with an even size, necessitating twice as many iterations for the same epoch.

**Evaluation.** We use the evaluation code provided by OpenCLIP. For zero-shot classification, we use the default template: "a photo of a [class]." We use this template for fashion datasets but use the template "an aerial photograph of a [class]" for remote sensing, following the setup of [64].

# D  Additional remote sensing results

## D.1  Different neural architectures

Table 11 presents the zero-shot classification results using the ResNet [99] and ViT [100] models. S-CLIP significantly improves the supervised baseline in all considered scenarios. Note that the larger model shows better overall accuracy, for both the original and fine-tuned CLIP. Here, S-CLIP gives reliable improvements when scaling up to larger models.

Table 11: Zero-shot classification results on remote sensing datasets using different neural architectures. Parentheses indicate the performance gap from the supervised CLIP, and bolds denote the best results within each architecture. S-CLIP give consistent improvements, scaling to larger networks.

| Model | Method | RSICD-CLS | UCM-CLS | WHU-RS19 | RSSCN7 | AID |
|---|---|---|---|---|---|---|
| ResNet-50 | CLIP (original) | 45.3 | 50.5 | 65.5 | 58.9 | 47.8 |
| | CLIP (fine-tune) | $58.3_{\pm0.3}$ | $63.5_{\pm3.4}$ | $76.5_{\pm3.2}$ | $61.9_{\pm1.2}$ | $63.1_{\pm1.3}$ |
| | S-CLIP (ours) | $\mathbf{66.9}_{\pm1.7}$ (+8.6) | $\mathbf{66.7}_{\pm1.6}$ (+3.2) | $\mathbf{86.9}_{\pm2.0}$ (+10.4) | $\mathbf{66.2}_{\pm1.1}$ (+4.3) | $\mathbf{73.0}_{\pm0.3}$ (+9.9) |
| ViT-B/32 | CLIP (original) | 55.8 | 58.6 | 76.4 | 62.1 | 55.7 |
| | CLIP (fine-tune) | $72.3_{\pm1.2}$ | $76.8_{\pm3.0}$ | $90.4_{\pm0.6}$ | $68.9_{\pm2.2}$ | $78.9_{\pm0.5}$ |
| | S-CLIP (ours) | $\mathbf{77.5}_{\pm1.0}$ (+5.2) | $\mathbf{78.3}_{\pm3.6}$ (+1.5) | $\mathbf{93.2}_{\pm1.9}$ (+2.8) | $\mathbf{69.6}_{\pm0.8}$ (+0.7) | $\mathbf{83.7}_{\pm1.8}$ (+4.8) |
| ViT-B/16 | CLIP (original) | 58.9 | 60.1 | 80.9 | 69.4 | 59.6 |
| | CLIP (fine-tune) | $72.4_{\pm0.6}$ | $78.7_{\pm2.7}$ | $90.0_{\pm1.4}$ | $71.9_{\pm3.8}$ | $77.9_{\pm0.9}$ |
| | S-CLIP (ours) | $\mathbf{79.5}_{\pm1.5}$ (+7.1) | $\mathbf{82.3}_{\pm0.4}$ (+3.6) | $\mathbf{93.9}_{\pm2.1}$ (+2.1) | $\mathbf{76.3}_{\pm0.3}$ (+4.4) | $\mathbf{85.2}_{\pm0.9}$ (+7.3) |

## D.2  Different image-text pair ratios

Table 12 presents the zero-shot classification results for different image-text pair ratios. Using only 10% of pairs (2× fewer than CLIP), S-CLIP achieves performance comparable to CLIP using 20% of pairs. Moreover, with 30% of pairs (3× fewer than CLIP), S-CLIP matches the performance of CLIP using 100% of pairs and even outperforms it, possibly due to the keyword-level pseudo-labeling.

Table 12: Zero-shot classification results on remote sensing datasets using different image-text pair ratios. Parentheses indicate the performance gap from the supervised CLIP, and bolds denote the best results within each pair ratio. S-CLIP gives consistent improvements, reducing required pairs.

| Method | Ratio | RSICD-CLS | UCM-CLS | WHU-RS19 | RSSCN7 | AID |
|---|---|---|---|---|---|---|
| CLIP (original) | 0% | 45.3 | 50.5 | 65.5 | 58.9 | 47.8 |
| CLIP (fine-tune) | 10% | $58.3_{\pm0.3}$ | $63.5_{\pm3.4}$ | $76.5_{\pm3.2}$ | $61.9_{\pm1.2}$ | $63.1_{\pm1.3}$ |
| S-CLIP (ours) | | $\mathbf{66.9}_{\pm1.7}$ (+8.6) | $\mathbf{66.7}_{\pm1.6}$ (+3.2) | $\mathbf{86.9}_{\pm2.0}$ (+10.4) | $\mathbf{66.2}_{\pm1.1}$ (+4.3) | $\mathbf{73.0}_{\pm0.3}$ (+9.9) |
| CLIP (fine-tune) | 20% | $66.2_{\pm1.1}$ | $69.7_{\pm0.2}$ | $82.8_{\pm1.5}$ | $65.6_{\pm3.4}$ | $71.8_{\pm1.1}$ |
| S-CLIP (ours) | | $\mathbf{77.6}_{\pm1.1}$ (+11.4) | $\mathbf{72.0}_{\pm0.7}$ (+2.3) | $\mathbf{90.7}_{\pm1.9}$ (+7.9) | $\mathbf{67.9}_{\pm2.6}$ (+2.3) | $\mathbf{83.7}_{\pm0.7}$ (+11.9) |
| CLIP (fine-tune) | 30% | $70.8_{\pm1.6}$ | $70.7_{\pm2.3}$ | $85.4_{\pm1.7}$ | $63.6_{\pm1.5}$ | $75.7_{\pm0.8}$ |
| S-CLIP (ours) | | $\mathbf{80.1}_{\pm1.7}$ (+9.3) | $\mathbf{74.0}_{\pm1.1}$ (+3.3) | $\mathbf{94.8}_{\pm1.2}$ (+9.4) | $\mathbf{69.6}_{\pm2.8}$ (+6.0) | $\mathbf{87.8}_{\pm0.5}$ (+12.1) |
| CLIP (fine-tune) | 100% | $77.5_{\pm1.5}$ | $76.5_{\pm0.5}$ | $93.9_{\pm0.9}$ | $71.2_{\pm0.3}$ | $83.6_{\pm1.6}$ |

### D.3 Using all image-text pairs

We used 10% of the image-text pairs as labeled data to compare two scenarios, L=U and L≠U, in the main paper. Having demonstrated the effectiveness of S-CLIP in the L≠U scenario, the next question is whether it would also be helpful when using all paired data and extra unlabeled data. Table 13 presents the results of using the entire RS-ALL dataset as labeled data and RESISC45 as the unlabeled data. S-CLIP is shown to be effective even in this challenging scenario.

Table 13: Zero-shot classification results on remote sensing datasets using all captioned images and extra unlabeled images, where the unlabeled data is from a different distribution than the labeled data. S-CLIP also provides improvements in this most challenging scenario.

| Model | Method | RSICD-CLS | UCM-CLS | WHU-RS19 | RSSCN7 | AID |
|---|---|---|---|---|---|---|
| ResNet-50 | CLIP (fine-tune) | $77.5_{\pm1.5}$ | $76.5_{\pm0.5}$ | $93.9_{\pm0.9}$ | $71.2_{\pm0.3}$ | $83.6_{\pm1.6}$ |
| | S-CLIP (ours) | $\mathbf{81.1}_{\pm0.2}$ (+3.6) | $\mathbf{77.7}_{\pm1.4}$ (+1.2) | $\mathbf{95.8}_{\pm0.9}$ (+1.9) | $\mathbf{72.3}_{\pm2.8}$ (+1.1) | $\mathbf{87.7}_{\pm0.4}$ (+4.1) |
| ViT-B/32 | CLIP (fine-tune) | $85.3_{\pm1.0}$ | $87.3_{\pm0.7}$ | $95.8_{\pm1.0}$ | $75.7_{\pm1.4}$ | $92.0_{\pm0.5}$ |
| | S-CLIP (ours) | $\mathbf{87.0}_{\pm0.6}$ (+1.7) | $\mathbf{88.6}_{\pm1.4}$ (+1.3) | $\mathbf{97.2}_{\pm0.4}$ (+1.4) | $\mathbf{76.2}_{\pm0.2}$ (+0.5) | $\mathbf{92.5}_{\pm0.8}$ (+0.5) |
| ViT-B/16 | CLIP (fine-tune) | $87.0_{\pm0.3}$ | $88.5_{\pm0.6}$ | $96.7_{\pm0.1}$ | $78.4_{\pm0.8}$ | $89.2_{\pm1.4}$ |
| | S-CLIP (ours) | $\mathbf{87.4}_{\pm1.2}$ (+0.4) | $\mathbf{88.9}_{\pm1.6}$ (+0.4) | $\mathbf{97.3}_{\pm0.4}$ (+0.6) | $\mathbf{79.1}_{\pm1.0}$ (+0.7) | $\mathbf{93.1}_{\pm1.4}$ (+3.9) |

### D.4 Image-text retrieval R@1

Table 14 presents the R@1 results of image-text retrieval, complementing Table 2 in the main paper which shows the R@5 results. S-CLIP outperforms the semi-supervised methods in most cases, except the supervised baseline works best for image→text retrieval on the UCM dataset.

Table 14: Image-text retrieval results on remote sensing datasets, following the same setup of Table 1. Bolds denote the best results among the semi-supervised methods within the same setups. S-CLIP performs the best in most cases, similar to the observation in Table 2.

| Method | Data | Image→text R@1 | | | Text→image R@1 | | |
|---|---|---|---|---|---|---|---|
| | | RSICD | UCM | Sydney | RSICD | UCM | Sydney |
| CLIP (original) | - | 2.1 | 7.1 | 10.3 | 2.2 | 7.1 | 20.7 |
| CLIP (fine-tune) | L | $3.6_{\pm0.2}$ | $10.6_{\pm2.1}$ | $13.2_{\pm2.0}$ | $3.7_{\pm0.3}$ | $10.0_{\pm0.5}$ | $14.4_{\pm2.0}$ |
| Hard-PL [16] | | $3.8_{\pm0.1}$ | $9.7_{\pm2.7}$ | $14.4_{\pm1.0}$ | $3.9_{\pm0.7}$ | $9.2_{\pm1.8}$ | $10.9_{\pm1.0}$ |
| Soft-PL [21] | L=U | $4.0_{\pm0.3}$ | $11.0_{\pm0.8}$ | $11.5_{\pm2.0}$ | $\mathbf{4.2}_{\pm0.4}$ | $9.8_{\pm1.5}$ | $16.1_{\pm4.3}$ |
| S-CLIP (ours) | | $\mathbf{4.2}_{\pm0.1}$ | $\mathbf{11.6}_{\pm1.1}$ | $\mathbf{14.9}_{\pm1.0}$ | $\mathbf{4.2}_{\pm0.6}$ | $\mathbf{11.1}_{\pm0.5}$ | $\mathbf{17.8}_{\pm1.0}$ |
| Hard-PL [16] | | $3.8_{\pm0.3}$ | $8.1_{\pm1.0}$ | $13.4_{\pm1.2}$ | $3.4_{\pm0.2}$ | $8.9_{\pm1.4}$ | $12.6_{\pm1.0}$ |
| Soft-PL [21] | L≠U | $3.8_{\pm0.4}$ | $8.4_{\pm1.2}$ | $10.9_{\pm1.0}$ | $3.5_{\pm0.3}$ | $9.4_{\pm1.0}$ | $14.9_{\pm1.0}$ |
| S-CLIP (ours) | | $\mathbf{4.2}_{\pm0.6}$ | $\mathbf{9.8}_{\pm0.6}$ | $\mathbf{13.8}_{\pm0.2}$ | $\mathbf{3.9}_{\pm1.1}$ | $\mathbf{10.8}_{\pm1.5}$ | $\mathbf{17.8}_{\pm5.0}$ |

# E Additional ablation studies

## E.1 General image domain (COCO) results

Table 15 presents results on the COCO [101] dataset. We observed that fine-tuning models using limited image-caption pairs degrades performance since the original CLIP already performs well. Therefore, we selected a subset of the "sports" category where the original CLIP performs relatively weakly. Following other setups, we used 10% of the data as labeled data and the rest as unlabeled data. We kept the training configuration consistent with our paper but ran 10 epochs to ensure model convergence. S-CLIP model outperforms others, even in this general image domain.

Table 15: Image-text retrieval results for the COCO sports category, where the original CLIP performs relatively weakly. S-CLIP also performs well in the general image domain.

| Method | Image-to-text retrieval | Text-to-image retrieval |
|---|---|---|
| CLIP (original) | 40.30 | 38.17 |
| CLIP (fine-tune) | 46.59 | 47.76 |
| Hard-PL | 48.72 | 47.12 |
| Soft-PL | 48.72 | 47.23 |
| S-CLIP (ours) | **49.79** | **48.40** |

## E.2 Additional pseudo-labeling baselines

Table 16 presents a comparison with pseudo-labeling baselines adapted from the sate-of-the-art semi-supervised image classification methods. Specifically, we employ the pseudo-labeling techniques of SemPPL [22] and RoPAWS [69]. SemPPL uses the mode of k-NN predictions. In our case, since captions are unique for each image, we assign a uniform probability over k-NN samples, which we refer to as "k-NN" pseudo-labeling. On the other hand, RoPAWS utilizes an analytic fixed-point solution for label propagation, which we refer to as "LabProp" pseudo-labeling. We used these baselines for caption-level pseudo-labeling and compare them with OT. We do not used keyword-level pseudo-labeling for a fair comparison. OT outperforms other pseudo-labeling approaches.

Table 16: Comparison with pseudo-labeling approaches adopted from the SOTA semi-supervised image classification methods, following the setup of Table 5. OT performs the best.

| Method | Zero-shot accuracy | Image-text retrieval |
|---|---|---|
| k-NN (k=5) | 61.1 | 8.8 |
| LabProp | 68.2 | 9.4 |
| OT (ours) | **69.6** | **10.5** |

## E.3 Qualitative examples of pseudo-labels

Figure 7 and Figure 8 provide qualitative examples of caption-level and keyword-level pseudo-labels, respectively. Optimal transport (OT) matches visually and semantically similar labeled images for unlabeled queries, providing meaningful caption-level pseudo-labels. In addition, the nearest labeled images share overlapping keywords, providing meaningful keyword-level pseudo-labels.

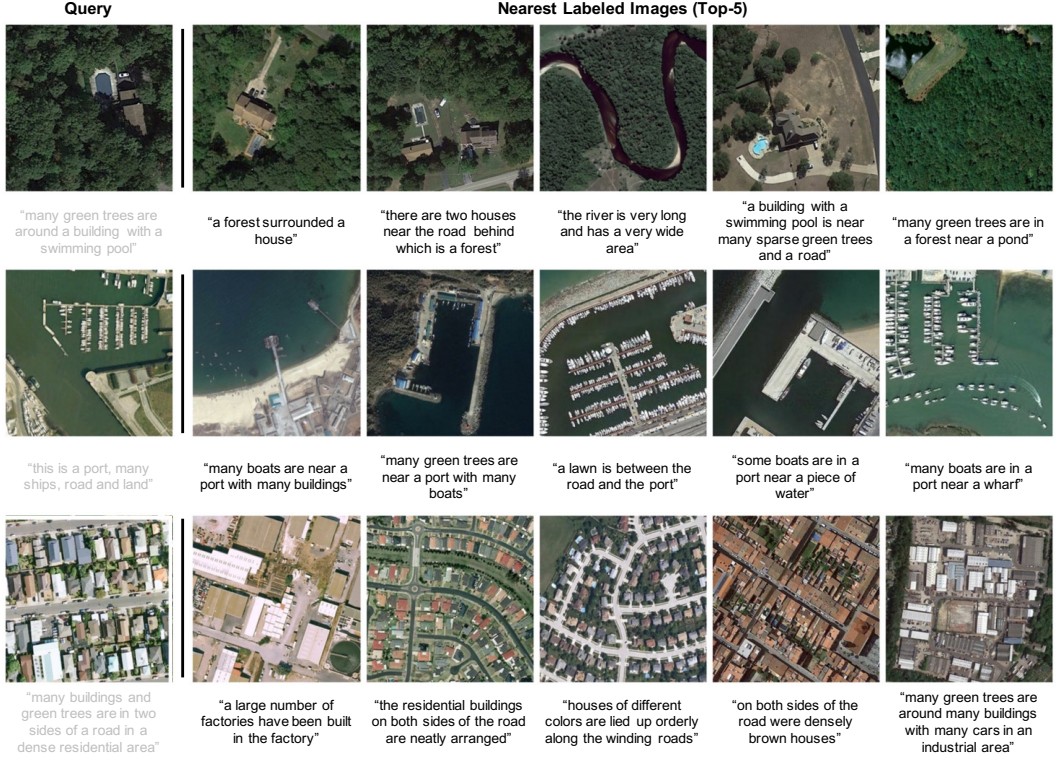

Figure 7: **Qualitative examples of caption-level pseudo-labels.** Optimal transport (OT) can identify visually and semantically similar images, including those representing forests, ports, and dense buildings. As a result, caption-level pseudo-labels offer accurate semantics for unlabeled images.

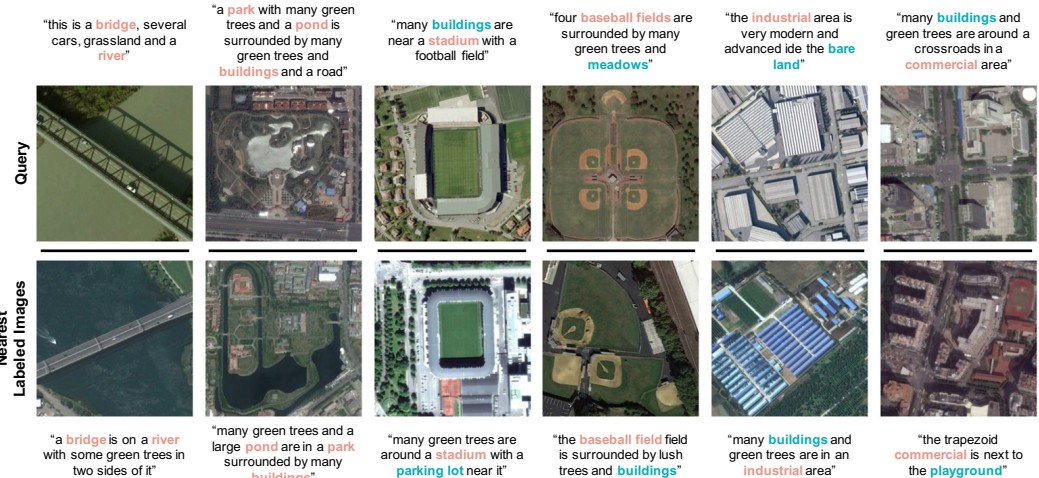

Figure 8: **Qualitative examples of keyword-level pseudo-labels.** The images closest to the query, as found by OT, often share the keywords in their captions (column 1-2). Even in cases where the images do not share all the keywords, they still have some overlapping keywords (column 3-6).

### E.4 Training curves

Figure 9 presents the training curves showing the loss and evaluation metrics. The validation accuracy for zero-shot classification on RSICD-CLS and image→text retrieval on RS-ALL is reported. The trends are similar in other datasets. Supervised CLIP is prone to overfitting, as indicated by the CLIP validation loss, but this is regularized by Hard-PL and Soft-PL. Our proposed S-CLIP yields the greatest benefits, with superior classification and retrieval performance.

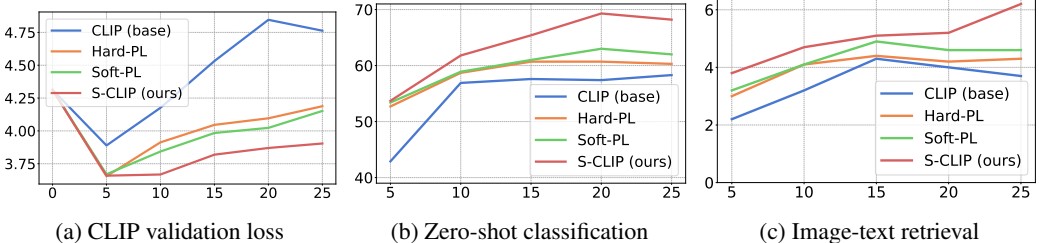

(a) CLIP validation loss      (b) Zero-shot classification      (c) Image-text retrieval

Figure 9: Trends in loss and evaluation metrics during training. S-CLIP regularizes overfitting, as shown by the CLIP validation loss. As a result, it performs the best classification and retrieval.

### E.5 Locked image or text tuning

Table 17 presents the results of fine-tuning after locking (freezing weights) either the image or text encoders, as suggested by LiT [41]. In our method, locking either encoder leads to a degradation in performance. This occurs because both the image and text data from the specialist domain are unseen, requiring the model to learn this novel information. In contrast, in the base CLIP, locking the text encoder yields similar results to having no lock.[6] This is due to the base CLIP suffering from overfitting and inadequately learning the text information. These results provide additional evidence that pseudo-labeling with S-CLIP improves the language understanding of models.

Table 17: Ablation study on locked image or text encoders. We report the average values of zero-shot accuracy (left) and image↔text retrieval (right). Locking either the image or text encoder degrades performance, as it prevents the model learning new information from specialist domains.

|  | Locked image | | Locked text | | No lock | |
|---|---|---|---|---|---|---|
| CLIP (fine-tune) | 48.7 | 8.6 | 64.8 | 9.4 | 64.7 | 9.3 |
| S-CLIP (ours) | 63.1 | 8.7 | 67.1 | 10.2 | **71.9** | **10.6** |

### E.6 Selection of mini-batches during training

Table 18 presents the effect of batch selection by applying different strategies to S-CLIP. Using a smaller batch size ($16 \times 4 = 64$) instead of the original ($64 \times 4 = 256$) significantly decreased the performance. However, collecting similar images helped mitigate this drop. We simplified the process by arranging batches based on their indices, sorted over visual similarities, and used them without shuffling. This method, "sorted batch," resulted in a reasonable gain. Exploring a more effective active batch sampling strategy would be an interesting future direction.

Table 18: Ablation study on the mini-batch sampling strategies, following the setup of Table 5. Using a larger and sorted batch improves performance by providing more informative pseudo-labels.

| Batch size | Zero-shot accuracy | Image-text retrieval |
|---|---|---|
| 64 (random batch) | 65.8 | 8.3 |
| 64 (sorted batch) | 67.3 | 8.9 |
| 256 (random batch) | **71.9** | **10.6** |

---

[6]The reported average values may appear similar, but each dataset has distinct values.

### E.7   Baselines using keyword information

Table 19 presents the results compared to the baselines using keyword information. Specifically, we incorporated two losses: (a) keyword loss on labeled data using multi-label classification, and (b) keyword loss on unlabeled data using our pseudo-labeling approach (nearest image is given by OT). To assess the impact of both losses, we compared CLIP-FT, CLIP-FT+(a), and CLIP-FT+(a)+(b). For the ablation study, we did not apply caption-level pseudo-labeling. Both (a) and (b) contribute to the overall performance, confirming that our proposed pseudo-labeling loss relies not only on keyword information but also on exploiting information from unlabeled data.

Table 19: Comparison to the baselines using keyword information, following the setup of Table 5. Both keyword information and semi-supervised learning contribute to the final performance.

| Method | Zero-shot accuracy | Image-text retrieval |
|---|---|---|
| CLIP-FT | 64.7 | 9.3 |
| + keyword info. | 67.3 | 9.7 |
| + semi-supervised | **70.6** | **10.1** |

### E.8   Keyword statistics in captions

We trained our models using captioned datasets, and the class names were derived from the classification counterparts of the dataset. Consequently, captions often contain multiple class names. For instance, 45.8% of captions in the RSICD dataset contain more than two class names. The remaining 50% of captions have only a single class name, which results in our keyword-level loss being reduced to a classification loss. It is worth noting that we developed a method to handle scenarios where class names are unavailable. By utilizing YAKE-200 keywords, we found that 94.8% of captions contain more than two keywords, confirming the necessity of partial label learning.

## F   Limitations and broader impacts

**Limitations.** Our pseudo-labeling approach is effective for zero-shot classification but less effective for image-text retrieval, which requires fine-grained language understanding. This is because the current pseudo-labeling approach assumes that captions in a batch can capture the semantics of unlabeled images, which may not hold true for fine-grained contexts. To address this issue, one can increase the batch size [84] or incorporate a queue of caption embeddings [85].

In addition, the current OT formulation of caption-level pseudo-labels assumes uniform sink and source constraints. However, this assumption can be refined in the case of distribution shifts. For example, techniques from robust semi-supervised learning [68, 69] can be utilized to estimate the in-domain-ness and subsequently regularize pseudo-labeling based on this estimation.

Lastly, keyword-level pseudo-labeling could be enhanced. The current method relies on the exact inclusion of keywords in the caption to form the candidate set. One can relax this criterion to include synonyms or related concepts by incorporating additional inference steps. In addition, the extracted keywords from captions may vary in their levels. Integrating a word hierarchy to account for different levels of keywords would be an interesting future research direction.

**Broader impacts.** Our paper aims to broaden the applicability of CLIP to specialist domains. However, this broader scope may present challenges when using the model with domains that include harmful content. These challenges arise from the data itself, rather than the model itself. Therefore, it is crucial to have adequate data regularization to effectively address these concerns.

