# OpenReview forum: "S-CLIP: Semi-supervised Vision-Language Learning using Few Specialist Captions"
_NeurIPS.cc/2023/Conference — NeurIPS 2023 poster_

### Official Review · Reviewer_Q2NM · 2023-07-06

**Soundness:** 3 good
**Presentation:** 4 excellent
**Contribution:** 3 good
**Rating:** 5
**Confidence:** 4

**Summary:**

The paper presents S-CLIP, a novel semi-supervised learning method for training contrastive language-image pre-training (CLIP) models in specialized domains where limited image-text pairs are available. Vision-language models like CLIP have achieved impressive results in natural image domains, but they struggle when applied to specialized domains like remote sensing due to the scarcity of paired data. To address this, S-CLIP leverages additional unpaired images and introduces two pseudo-labeling strategies: caption-level and keyword-level pseudo-labeling.

Caption-level pseudo-labeling assigns pseudo-labels to unlabeled images based on the captions of paired images. This is achieved by solving an optimal transport problem between the unlabeled and labeled images, ensuring that the pseudo-labels represent a probability distribution over the labeled images. On the other hand, keyword-level pseudo-labeling assigns pseudo-labels based on the keywords in the captions of visually similar paired images. It formulates the training as a partial label learning problem, considering a candidate set of target keywords instead of a single exact one.

The experiments conducted in various specialist domains, including remote sensing, fashion, scientific figures, and comics, demonstrate the effectiveness of S-CLIP. It outperforms CLIP fine-tuning and other semi-supervised learning competitors in zero-shot classification and image-text retrieval tasks. For example, in the remote sensing domain, S-CLIP improves zero-shot accuracy by 10.4% and image-to-text retrieval R@5 by 4.4% compared to CLIP fine-tuning. Moreover, S-CLIP remains robust even when the unlabeled images are from a different dataset. Overall, S-CLIP shows promise in enhancing CLIP training using only a few image-text pairs.

**Strengths:**

- Effective Pseudo-Labeling Strategies: The paper proposes two pseudo-labeling strategies, caption-level and keyword-level pseudo-labeling, which effectively guide the training process. The caption-level pseudo-labeling leverages optimal transport to assign pseudo-labels based on the captions of paired images, ensuring robust training even with distribution shifts. The keyword-level pseudo-labeling assigns pseudo-labels based on keywords in visually similar paired images, capturing local components of unlabeled images. The combination of these strategies results in improved zero-shot classification and image-text retrieval tasks.

- Experimental Demonstrations: The paper provides extensive experimental evaluations in various specialist domains, including remote sensing, fashion, scientific figures, and comics. The results consistently show that S-CLIP outperforms CLIP fine-tuning and other semi-supervised learning competitors, achieving significant improvements in zero-shot classification and image-text retrieval tasks. The robustness of S-CLIP, even when unlabeled images are from a different dataset, further strengthens its effectiveness.

- Relevance and Applicability: The paper addresses an important and practical problem in the field of vision-language models. The proposed S-CLIP method has the potential to enhance the training of CLIP models in specialized domains with limited paired data. The findings are relevant not only for researchers working on vision-language models but also for practitioners in various domains where image-text understanding is crucial.

- Clarity and Presentation: The paper is well-written, organized, and presents the proposed method and experimental results clearly. The figures and illustrations aid in understanding the concepts and methodologies. The strengths and weaknesses of the proposed method are discussed in a concise and informative manner.

**Weaknesses:**

- Baselines with other semi supervised approaches are severely limited with the added baselines being severely outdated. Several much newer few shot learning approaches are present and should be compared against. A summary could be found in the paper: A Survey on Deep Semi-supervised Learning by Yang et al. for example: SEMI-MAE: MASKED AUTOENCODERS FOR SEMI-SUPERVISED
VISION TRANSFORMERS by Yu et al or Semi-supervised Multimodal Representation Learning through a Global Workspace by Devillers et al
- A lot of the core content like limitations, experimental setup etc has been moved to the appendix which I feel is key to the understanding of the paper and unfairly grants the author extra writing space
- The approach combines keyword and caption level pseudo labelling, both of which have individually been well studied in literature. The combination of both is certainly a novel effort, but the contribution of the same doesnt seem like a strong enough novelty for a venue liek Neurips
- The choice of using OT as the distance measure between 2 images is interesting, it would be nice to see how other image distance measures compare to it

**Questions:**

- Could you clarify why the soft-PL, hard-PL were the only semi supervised baselines chosen for comparison despite the presence of a vast body of literature for semi supervised learning?

**Limitations:**

No ethical limitations as such

---

> ### Author Rebuttal · Authors · 2023-08-06
>
> Dear reviewer Q2NM,
>
> Thank you for your valuable feedback and comments. We appreciate your remarks on the strengths of our paper, including the problem significance, effective method, extensive experiments, and clear presentation. We will address your concerns and questions in the response below.
>
> ---
> **[W1] Comparison to other semi-supervised learning approaches**
>
> In our paper, we focus on developing a pseudo-labeling method for vision-language models, an underexplored research question in previous literature. As a result, we selected the two most representative pseudo-labeling approaches, Hard-PL and Soft-PL, as our baseline. These techniques can be integrated into existing semi-supervised learning frameworks, such as in combination with consistency regularization, as discussed in lines 100-102.
>
> For instance, your suggested Semi-MAE integrates Hard-PL alongside the MAE loss, and we can easily incorporate this into our method by applying the MAE loss to unlabeled images. On the other hand, your suggested global workspace (GW) necessitates a decoder for each modality, like image VAE and text BERT. However, obtaining these pre-trained models may not be feasible for the specialist domains we are focusing on. Considering that existing methods either complement our work or are not directly applicable to our specialized domain setup, we believe our baselines are not outdated.
>
> Nevertheless, for readers' further interest, we provide additional baselines that adapt the SOTA semi-supervised image classification methods. Specifically, we employ the pseudo-labeling technique of SemPPL [22] and RoPAWS [69], both published in ICLR 2023. SemPPL uses the mode of the k-NN prediction. In our case, since captions are unique for each image, we assign a uniform probability over k-NN samples, which we refer to as "k-NN" pseudo-labeling. On the other hand, RoPAWS utilizes an analytic fixed-point solution of label propagation, which we refer to as "LabProp" pseudo-labeling. We use these baselines for caption-label pseudo-labeling and compare them with OT. We do not apply keyword-level pseudo-labeling for ablation. The table below presents the results, following the setup of Table 5. OT outperforms other pseudo-labeling approaches.
>
>
> \begin{array}{lcc}
> \text{Method} & \text{Zero-shot accuracy} & \text{Image-text retrieval} \newline
> \hline
> \text{k-NN (k=5)} & 61.1 & 8.8 \newline
> \text{LabProp} & 68.2 & 9.4 \newline
> \text{OT (ours)} & 69.6 & 10.5 \newline
> \end{array}
>
> ---
> **[W2] Many contents are in the Appendix**
>
> We made our best efforts to convey our main ideas within the limited nine pages. As a result, we moved limitations and experimental setups to the Appendix, considering them less essential but valuable information for in-depth readers. We will include the mentioned contents in the final manuscript, where an additional content page is allowed.
>
> ---
> **[W3] Combination of known pseudo-labeling techniques**
>
> We note that the use of pseudo-labels for image-caption pairs was an underexplored research question. As a result, the concepts of caption-level and keyword-level pseudo-labeling were not well-studied in the literature. Our contribution lies in the design of these approaches rather than simply combining existing techniques. In the subsequent paragraphs, we recap the discussions on how our proposed approaches differ from the existing pseudo-labeling techniques for image classification.
>
> *Caption-level pseudo-labeling.* As we discussed in lines 631-636 in the Appendix, the OT-based approach, similar to our caption-level pseudo-labeling, was previously studied for semi-supervised image classification. However, we argue that the balancing effect of OT is more critical for vision-language models, given the inherent distribution shifts between labeled and unlabeled image captions. Our novel problem setup enhances the practical applicability of OT, which has been less recognized in the literature.
>
> *Keyword-level pseudo-labeling.* The need for partial label learning arises due to the possibility of a caption containing multiple keywords, which does not happen for image classification tasks. Our proposed technique is designed to address this unique problem in semi-supervised vision-language models, which is novel to our best knowledge.
>
> ---
> **[W4] Other distance measures than OT**
>
> We compared several distance measures to define pseudo-labels, including Soft-PL and OT, based on the embedding similarities between image-image and image-text. For instance, Table 5e illustrates that image-image similarity outperforms image-text similarity. To further demonstrate the impact of distance measures, we present additional results in the table below. We solely apply the caption-level pseudo-labeling for ablation. We follow the setup of Table 5 but use the L$\neq$U setup instead of the L=U setup to emphasize the robustness of image-image distance. The results confirm the significance of the proper choice of the distance measure. Exploring other distances would be an interesting future direction.
>
>
> \begin{array}{lcc}
> \text{Distance} & \text{Zero-shot accuracy} & \text{Image-text retrieval} \newline
> \hline
> \text{Soft-PL (image-text)} & 50.3 & 8.2 \newline
> \text{Soft-PL (image-image)} & 60.6 & 8.7 \newline
> \text{OT (image-text)} & 64.9 & 8.9 \newline
> \text{OT (image-image)} & 69.1 & 9.7 \newline
> \end{array}
>
> ---
> **[Q1] Why using Hard-PL and Soft-PL as baselines**
>
> As discussed in W1, our focus is on designing pseudo-labels for image-caption pairs, which are a core component of semi-supervised learning and are orthogonal to other parts such as self-supervision (or consistency regularization). Therefore, we have chosen two of the most representative pseudo-labeling techniques, Hard-PL and Soft-PL, as our baseline.
>
> ---
> Please let us know if you have any further concerns.
>
> Sincerely,\
> Authors

---

> > ### Comment · Reviewer_Q2NM · 2023-08-17
> >
> > I appreciate the authors feedback. I would like to keep my original positive rating :)

---

### Official Review · Reviewer_XrtM · 2023-07-06

**Soundness:** 3 good
**Presentation:** 3 good
**Contribution:** 3 good
**Rating:** 6
**Confidence:** 4

**Summary:**

The paper studies semi-supervised learning for contrastive vision-language pre-training, during which unlabelled/unpaired data is also accessible. The authors formulate the pseudo-labeling as an optimal transport (OT) problem and propose to use the Sinkhore algorithm to produce pseudo labels, which is validated to be superior to vanilla had/soft PL solution. Comparisons with other works in literature utilizing OT are properly discussed. Furthermore, the paper proposes to use the key-level pseudo label to account for partial label learning. Experiments on several data domains are conducted to validate its effectiveness.

**Strengths:**

- The paper is technically sound and easy to follow.
- The evaluation is self-sufficient with proper baselines, which helps to corroborate the effectiveness of the proposed loss.
- The ablative study is sufficient and showcases the impact of different composing parts of the method.
- Consistent performance gain over a variety of downstream datasets.

**Weaknesses:**

- For keyword-level pseudo-label, the compared baseline is somewhat not fair, since one could also leverage the keyword information for labeled/paired data in the vanilla fine-tuning setting, i.e., fine-tuning pre-trained CLIP with CLIP loss and multi-class classification loss. Additional experiments should be conducted to verify the gain derives from the semi-supervised part of it instead of simply leveraging the label information.
- It would also be nice to showcase results in general data (e.g., the ones evaluated in the CLIP paper, Flicker, etc.). Manual data split may be necessary but it helps to corroborate the generalizability of the proposed method.



**Questions:**

- The title is somewhat confusing. The paper studies semi-supervised fine-tuning on pre-trained CLIP weights, which is not suitable to phrase as "semi-supervised pre-training", where you are dealing with an unlabelled general domain.
- When keywords are obtained from class names, does one labeled image normally only contain just one keyword? If yes, does this reduce to a classification loss (since the target is one-hot)?


**Limitations:**

All limitations are properly addressed.

---

> ### Author Rebuttal · Authors · 2023-08-07
>
> Dear reviewer XrtM,
>
> Thank you for your valuable feedback and comments. We appreciate your remarks on the strengths of our paper, including the technical soundness, sufficient evaluation and ablation, and clear presentation. We will address your concerns and questions in the response below.
>
> ---
> **[W1] Baseline using the keyword information**
>
> Following your suggestion, we have included baseline results that leverage keyword information. Specifically, we incorporated two losses: (a) keyword loss on labeled data using multi-label classification, and (b) keyword loss on unlabeled data using our pseudo-labeling approach (nearest image is given by OT). To assess the impact of both losses, we compared CLIP-FT, CLIP-FT+(a), and CLIP-FT+(a)+(b). For the ablation study, we did not apply caption-level pseudo-labeling. The table below presents the results, following the setup of Table 5. Both (a) and (b) contribute to the overall performance, confirming that our proposed pseudo-labeling loss relies not only on keyword information but also on exploiting information from unlabeled data.
>
> \begin{array}{lcc}
> \text{Method} & \text{Zero-shot accuracy} & \text{Image-text retrieval} \newline
> \hline
> \text{CLIP-FT} & 64.7 & 9.3 \newline
> \text{+ keyword loss on L} & 67.3 & 9.7 \newline
> \text{+ keyword loss on L and U} & 70.6 & 10.1 \newline
> \end{array}
>
> ---
> **[W2] Experiments on the general image domain**
>
> We conducted additional experiments on the COCO dataset. Here, we observed that fine-tuning models using limited image-caption pairs degrades the performance, as the original CLIP already performs well. Thus, we used a subset of the "sports" category where the original CLIP performs relatively weakly. Following the setup described in our main paper, we used 10% of the data as labeled data and the rest as unlabeled data. We kept the training configuration consistent with our paper but ran 10 epochs to ensure model convergence. We report the image-text retrieval performance based on the R@5 metric. The table below shows that our proposed S-CLIP model outperforms others, even in this general image domain.
>
> \begin{array}{lcc}
> \text{Method} & \text{Image-to-text retrieval} & \text{Text-to-image retrieval} \newline
> \hline
> \text{CLIP (original)} & 40.30 & 38.17 \newline
> \text{CLIP (fine-tune)} & 46.59 & 47.76 \newline
> \text{Hard-PL} & 48.72 & 47.12 \newline
> \text{Soft-PL} & 48.72 & 47.23 \newline
> \text{S-CLIP (ours)} & 49.79 & 48.40 \newline
> \end{array}
>
> ---
> **[Q1] Usage of “pre-training” in the title**
>
> We used the term "pre-training" to highlight our goal of training a vision-language model capable of zero-shot classification and image-text retrieval. Although our approach can pre-train models from scratch, we fine-tuned CLIP to transfer its learned representations. Nevertheless, based on your suggestion, we will revise our paper title to use the term "learning" instead of "pre-training" to be less confusing.
>
> ---
> **[Q2] Number of keywords per caption**
>
> We trained our models using captioned datasets, while the class names were derived from the classification counterparts of the dataset. Consequently, captions often contain multiple class names. For instance, 45.8% of captions in the RSICD dataset contain more than two class names. However, we acknowledge that your point is fair; the remaining 50% of captions only have a single class name, resulting in our keyword-level loss reducing to a classification loss in such cases. It is worth noting that we developed a method to handle scenarios where class names are unavailable. By utilizing YAKE-200 keywords, we found that 94.8% of captions contain more than two keywords, thus confirming the necessity of partial label learning. We will incorporate this discussion in the final manuscript.
>
> ---
> Please let us know if you have any further concerns.
>
> Sincerely,\
> Authors

---

> > ### Comment · Reviewer_XrtM · 2023-08-19
> > **Response to Authors' Rebuttal**
> >
> > All my concerns are properly addressed. I will keep my original positive rating.

---

### Official Review · Reviewer_7J1f · 2023-07-07

**Soundness:** 3 good
**Presentation:** 3 good
**Contribution:** 3 good
**Rating:** 6
**Confidence:** 3

**Summary:**

The paper introduces an additional training method for vision-language models like contrastive language-image pre-training (CLIP). The proposed method consists of semi-supervised learning with two different pseudo labeling including caption- and keyword-level pserudo-label. These two pseudo labelings are complementarily understand image through different types of text representation. The experimental section describes the effectiveness of the proposed method in terms of zero-shot image recognition and image-text retrieval in several representative datasets. A relatively small image-text paired dataset can improve the zero-shot recognition and image-text retrieval with the proposed pseudo labelings.

**Strengths:**

+ The paper serves a method for semi-supervised learning on image-text paired datasets. Usually, it is quite difficult to construct the kind of dataset, however, the paper proposed that a simple yet effective idea to solve a problem in vision language pre-training like CLIP.

+ This paper also points out that limited focused areas such as remote sensing and fashion cannot be effectively used a vision-language models like CLIP as is. Therefore, the paper insists that an easily adaptive labeling approach and semi-supervised fine-tuning from pseudo labels are important for CLIP to adapt a specific image dataset.

+ In the experimental section, the proposed method S-CLIP is much better than that of used the original CLIP and fine-tuned CLIP (CLIP-FT). It is reasonable to assign soft labels rather than hard labels (Hard-PL) in this problem.

+ The paper is well-written and effectively communicates the main ideas in the paper, making it accessible to a broad audience. Moreover, there contains various visualizations, e.g., Figures 1 and 2, to confirm that the learnt representations of vision-language pre-training are meaningful.

**Weaknesses:**

- CLIP is pre-trained on image-text paired dataset widely extracted from the major space on the Internet. Therefore, it is understandable that the dataset collection does not work well in situations where only a small amount of paired data is collected, or where the paired data is not steadily available on the web. On the other hand, does the proposed dataset collection method only work on remote sensing, fashion, scientific figures, and comics? If you could show the vision-language application based on the proposed dataset collection in more situations, the paper must be more effective to broader readers. Probably, the approach could be shown in more generic situations, however, there are only 4 types of situations. Though the authors do not necessarilly add the experimental results, how about discussing the effectiveness for general purpose dataset collection?

**Questions:**

Please see above-mentioned comment in 'weaknesses'.

**Limitations:**

There are no negative limitations and societal impacts.

---

> ### Author Rebuttal · Authors · 2023-08-06
>
> Dear reviewer 7J1f,
>
> Thank you for your valuable feedback and comments. We appreciate your remarks on the strengths of our paper, including the problem significance, simple yet effective method, reasonable experiments, and clear presentation. We will address your concerns and questions in the response below.
>
> ---
> **[W1] Dataset collection beyond the considered four domains**
>
> We would like to clarify that our paper focuses on effective learning from a given limited set of image-text pairs, rather than on collecting such datasets. Our proposed semi-supervised learning method can be applied to any vision-language domain, and for our experiments, we have chosen four domains where we could access publicly available datasets.
>
> Nevertheless, we agree with your point that our work can achieve broader applicability across domains beyond the four we have chosen. Examples include medical imaging [R1], manufacturing visual anomalies [R2], or scientific images such as cells [R3]. For such domains, users may require caption annotations from human experts, such as doctors, factory workers, or scientists. We believe our semi-supervised learning framework would significantly reduce the annotation costs for collecting datasets from these specialists. We will include this discussion on dataset collection strategy in the final manuscript.
>
> [R1] https://physionet.org/content/mimic-cxr/2.0.0/ \
> [R2] https://www.mvtec.com/company/research/datasets/mvtec-ad \
> [R3] https://www.rxrx.ai/
>
> ---
> Please let us know if you have any further concerns.
>
> Sincerely,\
> Authors

---

> ### Comment · Reviewer_7J1f · 2023-08-21
> **Response to author rebuttal**
>
> Thank you so much for the authors. The rebuttal has addressed my question and concern. I will keep my paper rating.

---

### Official Review · Reviewer_CjR3 · 2023-07-08

**Soundness:** 4 excellent
**Presentation:** 4 excellent
**Contribution:** 4 excellent
**Rating:** 6
**Confidence:** 4

**Summary:**

Fine-tuning CLIP in specialized domains suffer the problem of limited available image-text pairs. This paper propose to address this challenge via semi-supervised learning methods with the additional unpaired images. Two pseudo-labeling strategies including the caption-level pseudo labeling and keyword-level pseudo labeling are proposed to construct valuable information about the unpaired images.
The problem addressed in the paper is meaningful and the method proposed in the article is demonstrated effective through comprehensive experiments.


**Strengths:**

(+) This article is well written, and the description of the method pipeline is clear and easy to understand.

(+) Experiments are sufficient.

(+) The problem it solves is meaningful.

**Weaknesses:**

(-) "Baseline" in table 5 is a little confusing. See question (1).

(-) The effectiveness of OT-based "Caption-level pseudo-label" compared with soft-PL should be further verified.



**Questions:**

(1) "Baseline" in table 5 is a little confusing. Is that refers to the "CLIP (fine-tune)" method or the "hard-PL" or "soft-PL" baselines ?

(2) It seems the role of the proposed OT-based "Caption-level pseudo-label" is similar as the soft-PL baseline. I suggest the authors provide a comparison of the effect of using the OT-based method and the soft-PL method on top of CLIP fine-tuned; or replace the OT-based method with the soft-PL, ablation on S-CLIP to show the effectiveness.

(3) I suggest the accuracy of training on 100% labels (full data) should be reported, as the upper bound performance to be reffered.

---

> ### Author Rebuttal · Authors · 2023-08-06
>
> Dear reviewer CjR3,
>
> Thank you for your valuable feedback and comments. We appreciate your remarks on the strengths of our paper, including the problem significance, sufficient experiments, and clear presentation. We will address your concerns and questions in the response below.
>
> ---
> **[W1] Definition of “Baseline” in Table 5**
>
> "Baseline" implies CLIP (fine-tune), since our goal was to ablate the effect of caption-level and keyword-level pseudo-labeling on top of this. We will clarify this in the final manuscript.
>
> ---
> **[W2] Advantage of OT over Soft-PL**
>
> OT becomes identical to Soft-PL when we set the number of Sinkhorn iterations to 0, as mentioned in line 304. The rationale behind this can be found in Appendix A.2, where the Sinkhorn algorithm is initialized as Soft-PL and progressively becomes balanced as the iterations proceed. In short, Soft-PL often assigns the pseudo-labels concentrated on certain captions, resulting in the collapse during training. OT regularizes this collapse, leading to the assignment of better pseudo-labels. The ablation study in Table 5b demonstrates the advantage of OT over Soft-PL with performance improvements over Sinkhorn iterations. We will further clarify this in the final manuscript.
>
> For your convenience, we have copied the results in the table below.
> \begin{array}{lcc}
> \text{Sinkhorn iteration} & \text{Zero-shot accuracy} & \text{Image-text retrieval} \newline
> \hline
> \text{Iter. 0 (= Soft-PL)} & 70.6 & 9.2 \newline
> \text{Iter. 1} & 71.4 & 9.4 \newline
> \text{Iter. 5} & 71.6 & 10.2 \newline
> \text{Iter. 10} & 71.9 & 10.6 \newline
> \end{array}
>
> ---
> **Q1** and **Q2** are answered in W1 and W2, respectively.
>
> ---
> **[Q3] Comparison with training using 100% labels**
>
> We provide a comparison of CLIP-FT and S-CLIP using different ratios of image-text pairs in Table 12 in Appendix E.2, including training on 100% of labels. We found that S-CLIP, using only 20% of labels, matches the performance of CLIP-FT trained on 100% of labels.
>
> For your convenience, we have copied the results in the table below.
>
> \begin{array}{lccc}
> \text{Method} & \text{Pair ratio (\\%)} & \text{RSICD-CLS} & \text{UCM-CLS} \newline
> \hline
> \text{CLIP (original)} & 0 & 45.3 & 50.5 \newline
> \hline
> \text{CLIP (fine-tune)} & 10 & 58.3 & 63.5 \newline
> \text{S-CLIP (ours)} & 10 & 66.9 & 66.7 \newline
> \hline
> \text{CLIP (fine-tune)} & 20 & 66.2 & 69.7 \newline
> \text{S-CLIP (ours)} & 20 & 77.6 & 72.0 \newline
> \hline
> \text{CLIP (fine-tune)} & 100 & 77.5 & 76.5 \newline
> \end{array}
>
> ---
> Please let us know if you have any further concerns.
>
> Sincerely,\
> Authors

---

> ### Comment · Reviewer_CjR3 · 2023-08-17
>
> I thank the authors for their response. My concerns have been adequately addressed. My opinion about the paper is still positive.

---

### Official Review · Reviewer_EbWV · 2023-07-09

**Soundness:** 3 good
**Presentation:** 3 good
**Contribution:** 3 good
**Rating:** 6
**Confidence:** 3

**Summary:**

This paper proposes a method S-CLIP, to enhance CLIP in a semi-supervised way by leveraging captions (via optimal transport) and keywords (via image-keyword similarity) from other unpaired images in the same batch. This is beneficial for specialized domains like remote sensing, where image-text data is usually limited. Results show better performance than the traditional pseudo-labelling methods, especially in OOD domains.

**Strengths:**

Addressing low-data regimes for contrastive learning. Such a problem is common in specialized domains like remote sensing.

Novel and simple method, but showing promising results compared to the standard pseudo-labelling methods, especially in OOD domains.

Experiments and ablations are comprehensive.

Well written and easy to read.


**Weaknesses:**

In section 5 experiments, missing experiment details for reproducing, such as model size, epoch, training schedule (lr, weight decay etc.)...

Besides the traditional pseudo-labelling methods, it would be nicer if we could compare with other SOTA pseudo-labelling methods for vision-language training – if there were no such SOTA methods, we could claim that in the paper.


**Questions:**

Given this method is sensitive to the batch data (e.g. statement in 5.2), it would be more solid if we could consider some data preprocessing strategies, e.g., to batch similar images in the same batch as much as possible? Or at least we should study how batch size affects the results in ablations?

In line 237, “we employ 32 image-caption pairs and 32 unpaired images”: this seems contrary with the problem to solve, i.e., N << M?

Could we give some qualitative examples for caption-level and keyword-level selection? E.g., given a few images, which are the selected text (weighted captions/keywords) look like in the same batch?

minor comment1: in Figure 1b, Hard-PL is “-2.7” instead of “-1.7”?

minor comment2: the entire section 3 background could be removed to save more space for ablations/analysis, which is more valuable to readers (given the background intro is too basic)?

minor comment3: explain what “YAKE-100/200/300” mean in Table 5?

minor comment4: “(f) Partial” to be consistent with other items.


**Limitations:**

No specific concern.

---

> ### Author Rebuttal · Authors · 2023-08-06
>
> Dear reviewer EbWV,
>
> Thank you for your valuable feedback and comments. We appreciate your remarks on the strengths of our paper, including the problem significance, novel and simple method, comprehensive experiments and ablations, and clear presentation. We will address your concerns and questions in the response below.
>
> ---
> **[W1] Experimental details**
>
> We have stated all the experimental details, such as epochs and training schedules, in Appendix D.2, as well as in our code provided in the supplementary material. We will clarify that line 240 of Section 5, stating “Additional details are in Appendix D.,” includes such information in the final manuscript.
>
> ---
> **[W2] Comparison to other pseudo-labeling methods**
>
> Designing pseudo-labeling methods for vision-language training is an underexplored research question. Specifically, our focus is on adapting CLIP for specialist domains, making prior approaches not directly applicable. For instance, prior works often require a pre-trained object detector to align detected objects with keywords in captions [7-9] or a pre-trained decoder to synthesize data in missing modalities [R1], which may not be suitable for specialist domains. Therefore, we believe that our work is SOTA for the problem we consider, and we will add this claim in the final manuscript as per your suggestion.
>
> Nevertheless, for readers' further interest, we provide additional baselines that adapt the SOTA semi-supervised image classification methods. Specifically, we employ the pseudo-labeling technique of SemPPL [22] and RoPAWS [69], both published in ICLR 2023. SemPPL uses the mode of the k-NN prediction. In our case, since captions are unique for each image, we assign a uniform probability over k-NN samples, which we refer to as "k-NN" pseudo-labeling. On the other hand, RoPAWS utilizes an analytic fixed-point solution of label propagation, which we refer to as "LabProp" pseudo-labeling. We use these baselines for caption-label pseudo-labeling and compare them with OT. We do not apply keyword-level pseudo-labeling for ablation. The table below presents the results, following the setup of Table 5. OT outperforms other pseudo-labeling approaches.
>
>
> \begin{array}{lcc}
> \text{Method} & \text{Zero-shot accuracy} & \text{Image-text retrieval} \newline
> \hline
> \text{k-NN (k=5)} & 61.1 & 8.8 \newline
> \text{LabProp} & 68.2 & 9.4 \newline
> \text{OT (ours)} & 69.6 & 10.5 \newline
> \end{array}
>
> [R1] Devillers et al. Semi-supervised Multimodal Representation Learning through a Global Workspace. arXiv, 2023/06/27.
>
> ---
> **[Q1] Selection of mini-batches during training**
>
> Thank you for the insightful suggestion! We conducted additional experiments to verify the effect of batch selection by applying different strategies to S-CLIP. The table below presents the results, following the setup of Table 5. Using a smaller batch size (16x4=64) instead of the original (64x4=256) significantly decreased the performance. However, collecting similar images helped mitigate this drop. We simplified the process by arranging batches based on their indices, sorted over visual similarities, and used them without shuffling. This method, "sorted batch," resulted in a reasonable gain. Exploring a more effective active batch sampling strategy would be an interesting future direction.
>
>
> \begin{array}{lcc}
> \text{Batch size} & \text{Zero-shot accuracy} & \text{Image-text retrieval} \newline
> \hline
> \text{64 (random batch)} & 65.8 & 8.3 \newline
> \text{64 (sorted batch)} & 67.3 & 8.9 \newline
> \text{256 (random batch)} & 71.9 & 10.6 \newline
> \end{array}
>
> ---
> **[Q2] Size of the labeled vs. unlabeled data**
>
> We used 32 image-caption pairs and 32 unpaired images for each batch per GPU, where N and M indicate the total numbers of paired and unpaired images, respectively. In our experiments, N << M; for instance, M is 9 times larger than N when we subsample 10% of images as paired ones. We will further clarify in the final manuscript that 32 implies the batch size, not the dataset size, as also mentioned on line 236.
>
> ---
> **[Q3] Qualitative examples of pseudo-labels**
>
> We provide the qualitative examples of pseudo-labels in the attached PDF in the general response. The results demonstrate that optimal transport (OT) matches visually and semantically similar labeled images for unlabeled queries, providing meaningful caption-level pseudo-labels. In addition, the nearest labeled images share overlapping keywords, providing meaningful keyword-level pseudo-labels.
>
> ---
> **Minor comments**
>
> 1, 4: Thank you for the clarification. We will correct the typos in the final manuscript.\
> 2: We included the background section for self-containment. However, we will consider relocating it to the Appendix in the final manuscript.\
> 3: YAKE-N implies that we utilize the top-N keywords from the YAKE algorithm. We will provide further clarification on this in the final manuscript.
>
> ---
> Please let us know if you have any further concerns.
>
> Sincerely,\
> Authors

---

> > ### Comment · Reviewer_EbWV · 2023-08-17
> >
> > Thanks for the clear and detailed response, which has generally addressed all my concerns. I would like to keep my original positive rating.

---

### Author Rebuttal · Authors · 2023-08-06

To answer the Q3 of reviewer EbWV, we have attached a PDF containing qualitative examples of caption-level and keyword-level pseudo-labels.

---

### Decision · Program_Chairs · 2023-09-21

**Decision:**

Accept (poster)

**Comment:**

This paper proposes to enhance CLIP in a semi-supervised way with the help of captions. The proposed approach could benefit domains where image-text data are difficult to collect.

Concerns were raised mainly regarding the implementation details and baseline comparisons. The authors provided a rebuttal and after the discussion period, all reviewers vote for acceptance. The AC recommends acceptance, and encourages the authors to revise paper accordingly.